# Molecular basis of ligand binding and receptor activation at the human A₃ adenosine receptor

Liudi Zhang[1,2,9], Jesse I. Mobbs [1,2,9], Felix M. Bennetts [1,2], Hariprasad Venugopal [3], Anh T. N. Nguyen[1], Arthur Christopoulos [1,2,4], Daan van der Es [5], Laura H. Heitman [5,6], Lauren T. May [1] ✉, Alisa Glukhova [1,2,7,8] ✉ & David M. Thal [1,2] ✉

Adenosine receptors (ARs: A₁AR, A₂ₐAR, A₂ʙAR, and A₃AR) are crucial therapeutic targets; however, developing selective, efficacious drugs for them remains a significant challenge. Here, we present high-resolution cryo-electron microscopy (cryo-EM) structures of the human A₃AR in three distinct functional states: bound to the endogenous agonist adenosine, the clinically relevant agonist Piclidenoson, and the covalent antagonist LUF7602. These structures, complemented by mutagenesis and pharmacological studies, reveal an A₃AR activation mechanism that involves an extensive hydrogen bond network from the extracellular surface down to the orthosteric binding site. In addition, we identify a cryptic pocket that accommodates the N⁶-iodobenzyl group of Piclidenoson through a ligand-dependent conformational change of M174$^{5.35}$. Our comprehensive structural and functional characterisation of A₃AR advances our understanding of adenosine receptor pharmacology and establishes a foundation for developing more selective therapeutics for various disorders, including inflammatory diseases, cancer, and glaucoma.

The four adenosine receptors (ARs: A₁AR, A₂ₐAR, A₂ʙAR, and A₃AR) are Class A G protein-coupled receptors (GPCRs) activated by extracellular levels of the nucleoside adenosine[1]. These receptors are widely expressed in humans and regulate a diverse range of physiological processes. Tremendous effort has gone into modulating the activity of adenosine receptors as potential treatments for cardiovascular disease, nervous system disorders, inflammation, renal and endocrine disorders, cancer, and visual disorders[1–3]. In this regard, the human A₃AR is enigmatic because it plays dual roles

under different pathophysiological conditions[4,5]. This complexity is particularly evident in cancer biology. For example, A₃AR is overexpressed in several types of tumours and is a proposed diagnostic marker[6–9]. A₃AR overexpression suggests a pro-tumoral role, promoting cell proliferation and survival[10–12]. In contrast, in other cancer types, the activation of A₃AR demonstrated anti-tumoral effects by triggering apoptosis and inhibiting cell growth[13–15]. Nevertheless, the A₃AR selective agonists Piclidenoson and Namodenoson have progressed into clinical trials for treating inflammatory diseases,

¹Drug Discovery Biology, Monash Institute of Pharmaceutical Sciences, Monash University, Parkville, VIC, Australia. ²ARC Centre for Cryo-Electron Microscopy of Membrane Proteins, Monash Institute of Pharmaceutical Sciences, Monash University, Parkville, VIC, Australia. ³Ramaciotti Centre for Cryo-Electron Microscopy, Monash University, Clayton, VIC, Australia. ⁴Neuromedicines Discovery Centre, Monash University, Parkville, VIC, Australia. ⁵Division of Medicinal Chemistry, Leiden Academic Centre for Drug Research, Leiden University, Leiden, The Netherlands. ⁶Oncode Institute, Leiden, The Netherlands. ⁷The Walter and Eliza Hall Institute of Medical Research, Parkville, VIC, Australia. ⁸Department of Biochemistry and Pharmacology, The University of Melbourne, Melbourne, VIC, Australia. ⁹These authors contributed equally: Liudi Zhang, Jesse I. Mobbs. ✉e-mail: lauren.may@monash.edu; glukhova.a@wehi.edu.au; david.thal@monash.edu

including rheumatoid arthritis, psoriasis, and liver diseases such as hepatocellular carcinoma, hepatitis, and dry eye syndrome[16–22]. In contrast, A$_3$AR antagonists are being developed as treatments for glaucoma and asthma[23–26].

Despite the vast therapeutic potential of adenosine receptors, few drug candidates have progressed through to the clinic. A major challenge in GPCR drug discovery, particularly relevant to adenosine receptors, is identifying ligands that selectively target one receptor subtype over similarly related subtypes; lack of such selectivity can lead to off-target side effects. To address this challenge, structural studies of GPCRs with various ligands have begun to illuminate molecular mechanisms of ligand selectivity, or lack thereof, paving the way for a new era of drug discovery.

Regarding adenosine receptors, the A$_{2A}$AR was a model GPCR for pioneering structural biology work[27] with various antagonist, agonist, and partial agonist-bound structures available, including the receptor in inactive[28], intermediate[29,30], and fully active conformations[31]. Structures of the A$_1$AR, A$_{2A}$AR, and A$_{2B}$AR have revealed key principles of adenosine receptor activation and selectivity that have guided drug design efforts[32–36]. Recent experimental structures of the A$_3$AR have been determined with the agonists Piclidenoson and Namodenoson[37,38]. However, crucially, these structures are relatively low resolution, and important parts of the ligand, such as an N[6]-iodobenzyl group, were not adequately resolved in the structures. Moreover, the structures of the A$_3$AR in both inactive and active conformations are crucial for elucidating its activation mechanism and guiding the design of future subtype-selective ligands.

In this study, we determine a Fab-assisted[39,40] cryo-EM structure of the human A$_3$AR in the inactive conformation, to a resolution of 2.8 Å, in complex with the covalent antagonist LUF7602[41]. In addition, we report cryo-EM structures of the human WT A$_3$AR in complex with the G protein and bound to the endogenous agonist adenosine and the clinically relevant agonist Piclidenoson. Our A$_3$AR structures, combined with pharmacological data, provide insights into the ligand-binding modes of both antagonists and agonists. Comparison of the inactive and active conformations of the human A$_3$AR revealed a molecular basis for receptor activation. This activation is mediated by an extensive hydrogen bond network that extends from the top of TM7 through TM1, TM2, and TM3 down to the core of the orthosteric binding site. These findings not only enhance our understanding of A$_3$AR ligand binding, activation, and signalling mechanisms but also provide a robust structural framework for the rational design of highly selective A$_3$AR ligands that could lead to new treatment strategies for a wide range of disorders, including inflammatory diseases, cancer, and glaucoma.

## Results

### Structures of the inactive conformations of the human A$_3$AR

Expression of human A$_3$AR was enhanced by inserting the first 22 amino acids from the human M$_4$ muscarinic receptor (M$_4$ mAChR) between an N-terminal FLAG epitope and the wild-type (WT) human A$_3$AR sequence (Fig. S1A)[32]. The pharmacological behaviour of the A$_3$AR construct was assessed using a [35S]GTPγS binding assay and showed minimal differences in receptor activation mediated by the agonist NECA compared to WT A$_3$AR (Fig. S1B). To determine the inactive state structure of the A$_3$AR using cryo-EM, we introduced several modifications to improve protein stability and expression (Fig. S2A). The intracellular loop 3 (ICL3) was replaced with BRIL, and the S97R[3.39] mutation was introduced to stabilise the inactive conformation[42–44]. Additionally, we removed a potential glycosylation site in ECL2 (N160A) to reduce conformational heterogeneity[45]. Residue numbering follows the numbering scheme by Ballesteros and Weinstein[46] and the GPCRdb numbering scheme for the ECL regions[47].

Purified A$_3$AR-BRIL-S97R was incubated with an anti-BRIL Fab fragment (BAG2) and an anti-BAG2 nanobody (Nb) to facilitate

structure determination, serving as fiducial markers and increasing the molecular weight of the complex (Fig. S2B–F)[39,40]. We initially attempted to determine the A$_3$AR structure with the antagonist MRS1220[48]. The global resolution reached 3.7 Å, with well-defined regions for the Fab-Nb-BRIL complex and lower TM segments of A$_3$AR (Fig. S3). However, no cryo-EM density was observed in the orthosteric site, suggesting the structure was either ligand-free or had low ligand occupancy (Fig. S3E, F). Additionally, the cryo-EM density corresponding to ECL2 was too ambiguous for accurate modelling, though ECL1 and ECL3 backbones were traceable (Fig. S3D). To improve resolution, we used the covalent antagonist LUF7602 (Fig. 1A), which is a covalent antagonist that was designed based on a high-affinity tricyclic xanthine scaffold[49] and the A$_1$AR covalent antagonist FSCPX[50].

A NanoBRET binding assay[51,52] using an N-terminal NanoLuc (Nluc-A$_3$AR) tagged receptor and the fluorescent antagonist XAC analogue (XAC-630)[53] confirmed that LUF7602 exhibited wash-resistant inhibition with consistent binding affinity across A$_3$AR-WT, A$_3$AR-BRIL, and A$_3$AR-BRIL-S97R constructs (Fig. S2G–H)[41,54]. We determined the structure of the A$_3$AR-LUF7602 complex at a global resolution of 2.7 Å (Figs. 1B and S4 and Table S1). Focused refinement further improved the quality of the receptor density, yielding a map at 3.3 Å (Figs. 1C and S4), enabling clear assignment of most receptor residues except for those near the covalent sulfonyl group attachment and Y265[7.36] (Figs. 1C and S5A). We note that we did not observe a clear EM density for the covalent linkage between the benzene-sulfonate group of LUF7602 and Y265[7.36]. We assigned the covalent attachment based on our pharmacology experiments showing irreversible binding (Fig. S2G), a prior study identifying Y265[7.36] as the point of covalent attachment[41], and similarity to the A$_1$AR bound to the irreversible antagonist DU172 (PDB: 5UEN)[32]. Poor EM density around the benzene-sulfonate group, Y265[7.36], and Y15[1.35] (Fig. 2A) suggests conformational heterogeneity potentially due to the covalent attachment not being 100% complete, consistent with ~15% rebinding of XAC-630 in the washout experiments (Fig. S2G).

### Comparison of the inactive conformation of adenosine receptors

The overall inactive conformation of LUF7602-bound A$_3$AR shares similarities with other adenosine receptor structures bound to xanthine-based antagonists, including structures of the A$_{2A}$AR bound to caffeine (PDB: 5MZP), theophylline (PDB: 5MZJ), XAC (PDB: 3REY), Istradefylline (PDB: 8GNG)[33,55,56], and the A$_1$AR bound to the irreversible antagonist DU172 (PDB: 5UEN) and PSB36 (PDB: 5N2S) (Fig. 1D)[32,33]. Despite relatively low sequence similarity, globally, the inactive conformation of the A$_1$AR, A$_{2A}$AR, and A$_3$AR receptors align well with RMSD values of less than 1 Å across the TM regions (Fig. 1D–J). Regions of high similarity include the positions of the TM helices (Fig. 1D), the intracellular loops (ICLs) (Fig. 1F), and the C-terminal region of ECL2, which is constrained due to a conserved disulfide bond (D1), C83[3.25] and C166[ECL2], that is common in Class A GPCRs (Fig. 1H).

Divergence in the inactive conformation of adenosine receptor structures arises in the extracellular regions of the TM helices and ECLs (Fig. 1E–J). These regions have the lowest sequence similarity across subtypes, with ECL2 and ECL3 varying in length. Indeed, the ECLs have been demonstrated to play a pivotal role in facilitating ligand entry and binding at adenosine receptors and other GPCRs[27,57–63]. The conformation of ECL1 was similar across adenosine receptor subtypes, with the C-terminal region of ECL1 forming interactions that stabilise the conformation of ECL2 (Fig. 1G). There is more divergence near the TM2 region of ECL2. For example, compared to the A$_{2A}$AR, there was a 5 Å outward displacement of TM2 in the DU172-bound A$_1$AR structure due to the proximity of the benzene-sulfonate linkage that covalently links DU172 to TM7. A similar 3 Å TM2 shift was observed in the LUF7602-bound A$_3$AR structure. Interestingly, a comparison with our MRS1220-A$_3$AR dataset revealed an 8 Å outward shift of TM2. However,

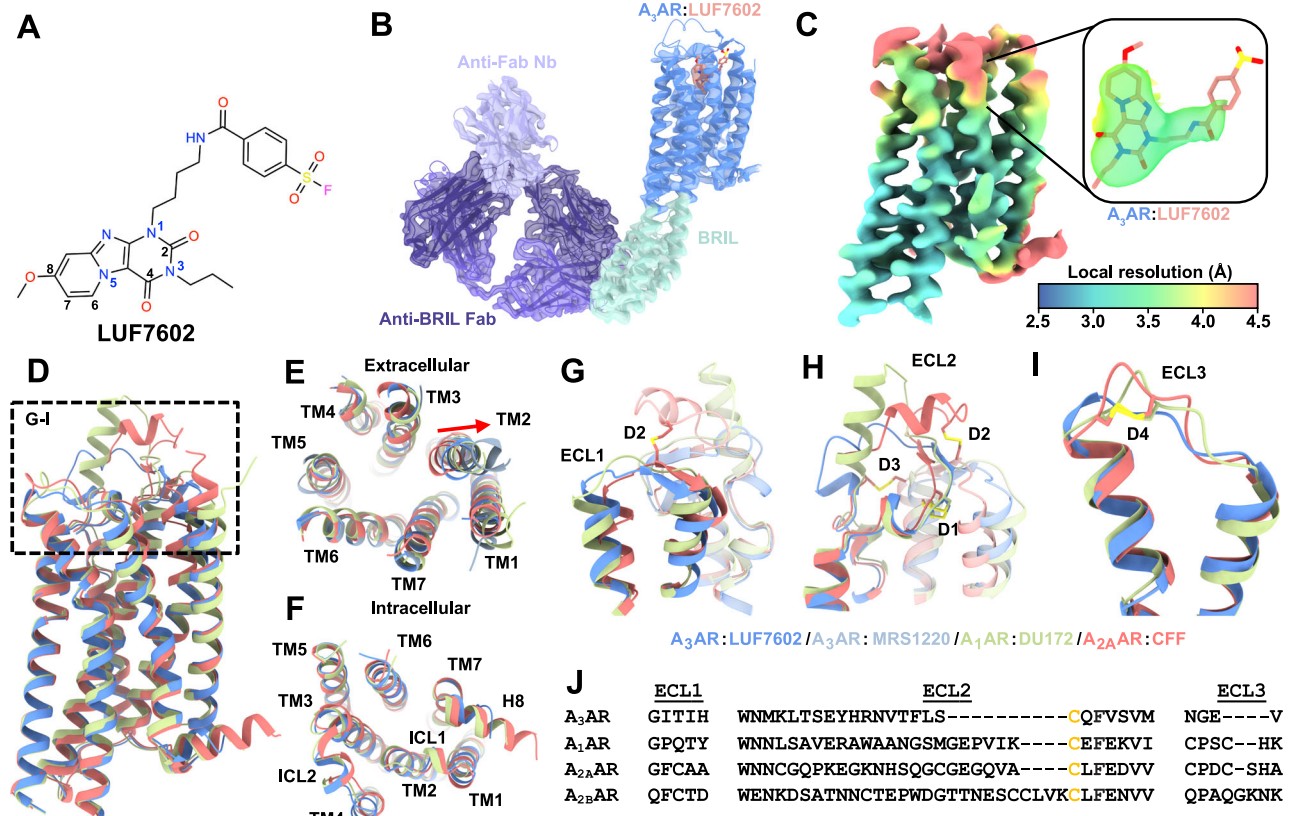

**Fig. 1 | Cryo-EM structure of the A₃AR bound to LUF7602 and comparison of the inactive conformation of adenosine receptors. A** Chemical structure of LUF7602 with key atoms numbered. **B** Cryo-EM density of consensus map (contour = 0.25) of the full inactive A₃AR:LUF7602 complex, showing the receptor (blue), LUF7602 (peach), BRIL fusion protein (light green), anti-Fab Nb (light purple), and anti-BRIL Fab (dark purple). **C** Local resolution receptor-focused cryo-EM map (contour = 0.23) with inset of cryo-EM density around LUF7602. Coloured by local resolution. **D** Overall structural alignment of inactive A₃AR:LUF7602 (blue), A₁AR:DU172 (PDB: 5UEN, light green), and A₂ₐAR:CFF (PDB: 5MZP, peach). **E** Extracellular and **F** intracellular views of the aligned structures, showing transmembrane helices (TM1-TM7) and helix 8 (H8). **G–I** Detailed views of extracellular loops **G** ECL1, **H** ECL2, **I** ECL3, and associated helices with disulfide bonds (D1–D4) are shown as sticks. **J** Sequence alignment of ECL1, ECL2, and ECL3 for the adenosine receptors.

the overall significance of this is caveated by the low resolution of the structure (Fig. 1E, light blue). Nevertheless, the different conformations of TM2 and ECL1 highlight their role in ligand binding.

The structure of ECL2 varies across AR subtypes, with differences in the number of disulfide bonds in each AR subtype affecting the tertiary loop structure (with 1 ECL2 disulfide bond in the A₁AR and A₃AR, 2 in A₂ᵦAR, and 3 in A₂ₐAR) (Fig. 1H). In the A₁AR, ECL2 forms a 3-turn α-helix that extends perpendicular to the membrane and then loops down to form an anti-parallel β-strand with ECL1, extending over the orthosteric binding site towards TM5 (Fig. 1H). In contrast, even though the length of ECL2 in the A₂ₐAR is similar to the A₁AR, the loop coils upward, forming a disulfide bond with TM3, followed by a 2.5-turn α-helix parallel to the membrane that also forms a disulfide bond with ECL1 before looping across the orthosteric site. The A₃AR has the shortest ECL2, with a small helix parallel to the membrane, before looping over to an anti-parallel β-strand with ECL1, followed by the conserved α-helix over the orthosteric site. The ECL2 of the A₂ᵦAR is the longest in sequence, but structural information for this loop is lacking. The conformation of ECL3 appears to be similar across adenosine receptor subtypes, except for the A₃AR, which lacks an initial loop following TM6 due to the lack of a disulfide bond and a shorter sequence (Fig. 1I).

**Ligand interactions at the inactive A₃AR**
Clear EM density revealed LUF7602 deeply bound in the A₃AR orthosteric site, interacting with TM helices 1–3, 5–7, and ECL2 (Fig. 2A, B). The xanthine core of LUF7602 formed a π–π stacking

interaction with F168[45.52], a common adenosine receptor interaction (Fig. 2C). To probe the pharmacology of the A₃AR orthosteric binding site and validate observed interactions from the ligand-bound A₃AR structures, we used a saturation binding NanoBRET assay to measure the binding affinity of the fluorescent antagonist XAC-630 to WT and mutant A₃AR constructs (Fig. 2G, H and Table 1). Mutations were based on residues interacting with LUF7602, adenosine, and Piclidenoson. XAC-630 showed no specific binding to the N250A[6.55] and H272A7[7.43] mutants, consistent with previous findings[64]. Overall, the point mutations had minimal effect on the binding affinity of XAC-630, with the H95A[3.37] and H95F[3.37] mutations reducing affinity ~5-fold and the Y15A[1.35] and M174A[5.35] mutations reducing affinity 3-fold. However, multiple mutations significantly decreased $B_{max}$ values compared to WT A₃AR, indicating these residues lowered receptor expression (Fig. 2G and Table 1).

We then assessed the ability of LUF7602 to compete with XAC-630 using a competition-binding NanoBRET assay (Fig. 2I, J and Table 2). The Y265A[7.36] mutant reduced LUF7602 affinity 15-fold, consistent with previously reported values[41]. Nearby residue Y15[1.35] forms a hydrogen bond with the amine group that links the xanthine core of LUF7602 to the reactive benzene-sulfonate warhead (Fig. 2D). Loss of the hydrogen bond interaction via Y15A[1.35] reduced LUF7602 affinity 10-fold. Residue N250[6.55] is a conserved residue that forms hydrogen bonds with the heterocyclic rings of adenosine receptor agonists and antagonists. In the LUF7602-bound A₃AR structure, N250[6.55] forms a single hydrogen bond with the carbonyl oxygen from the C⁴-position of the xanthine core (Fig. 2E, F). We could not assess the impact of

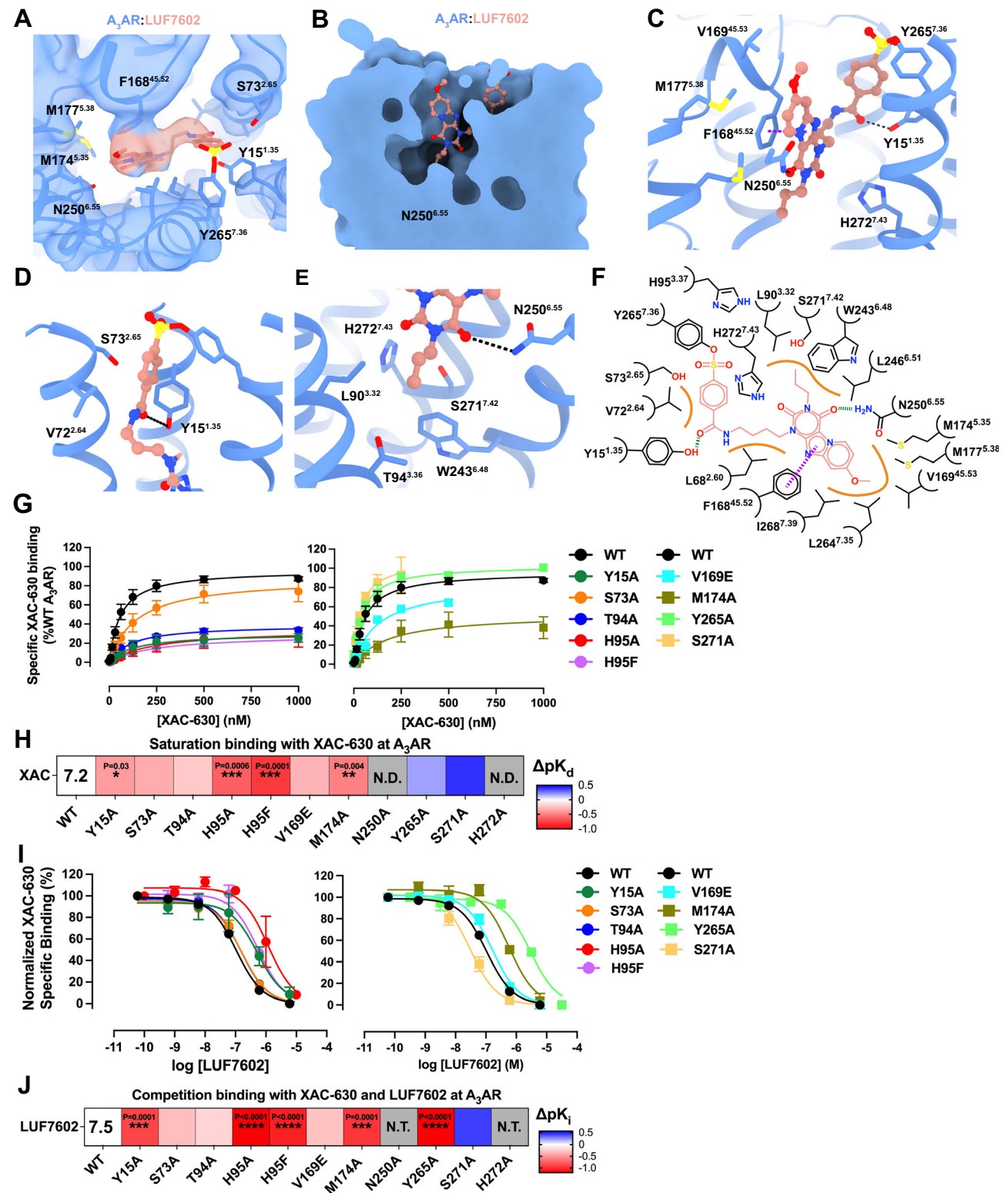

mutating N250$^{6.55}$ due to a loss of XAC-630 binding, highlighting its general importance. The remaining interactions of LUF7602 with the A$_3$AR were primarily hydrophobic interactions (Fig. 2F). The H95A$^{3.37}$ and H95F$^{3.37}$ mutations disrupted LUF7602 binding ~10-fold, consistent with H95 being an important residue for the binding of agonists and antagonists[64]. Similarly, the M174A$^{5.35}$ mutant decreased LUF7602 affinity 10-fold (Fig. 2I, J and Table 2). Finally, the affinity of LUF7602 for the S73 A$^{2.65}$ and T94A$^{3.36}$ mutants was similar to WT A$_3$AR, while the S271A$^{7.42}$ caused a slight increase in affinity, likely due to the removal of

a polar interaction and replacement with a hydrophobic interaction. Overall, these data support the binding mode of LUF7602 being due to diverse receptor-ligand interactions and provide opportunities to facilitate the design of higher-affinity A$_3$AR antagonists.

Comparison of xanthine-based antagonists bound to the orthosteric binding site of the A$_1$AR, A$_{2A}$AR, and A$_3$AR revealed common and distinct binding interactions (Fig. 3). Overall, the xanthine scaffolds occupy a similar pocket and form common interactions that include a π−π stacking interaction with conserved residue F$^{45.52}$ and a hydrogen

**Fig. 2 | LUF7602 binding site at the A3AR. A** Cryo-EM density of the receptor-focused map of A$_3$AR (blue mesh, contour = 0.23) with bound LUF7602 (peach). **B** Surface representation of the A$_3$AR orthosteric binding pocket with LUF7602 (peach sticks). **C** View of key residue interactions with LUF7602 in the binding pocket. **D** Detailed view of the covalent attachment of LUF7602 to Y265[7.36] and hydrogen bond with Y15[1.35]. **E** Hydrogen bond interactions between LUF7602 with N250[6.55]. **F** Schematic representation of LUF7602 interactions within the A$_3$AR binding pocket. Hydrogen bonds are shown as green dashed lines, a π-π stacking interaction as purple dashed lines, and orange lines as hydrophobic interactions. **G** NanoBRET saturation specific binding curves for XAC-630 at WT A$_3$AR and various mutants. Maximal specific binding was normalised to 100% of WT A$_3$AR. Data shown are grouped with mean ± SEM values. Grouped pK$_d$ values and statistical analysis were from $n = 6$ (WT), $n = 5$ (Y15A, M174A), $n = 4$ (H95A), and $n = 3$ (S73A, T94A, H95F, V169E, N250A, Y265A, S271A, and H272A). pK$_d$ and $B_{max}$ values are reported in Table 1. **H** Heat map showing changes in binding affinity (ΔpK$_d$) of XAC-630 to A$_3$AR mutants relative to WT in saturation binding assays. N.D. not determined due to no measurable response. The pK$_d$ for XAC-630 at WT is overlaid on the heatmap. **I** Competitive binding curves showing the displacement of XAC-630 by LUF7602 at WT A$_3$AR and selected mutants. The concentration of XAC-630 used in these experiments was approximately the K$_d$ determined in (**G**), with 200 nM used for all A$_3$AR constructs except S271A, where 40 nM was used. Grouped data are shown as mean ± SEM from $n = 3$ experiments ($n = 4$ for WT and Y15A) performed in duplicate. Grouped pK$_i$ values and statistical analysis are shown in Table 2. **J** Heat map showing changes in binding affinity (ΔpK$_i$) of LUF7602 competing with XAC-630 at A$_3$AR mutants relative to WT. The pK$_i$ for LUF7602 at WT is overlaid on the heatmap. **H, J** Significant differences vs WT were determined by one-way ANOVA with a Dunnett's multiple comparison test. *P* values: * = 0.01–0.05; ** = 0.001–0.01; *** = 0.0001–0.001; **** = <0.0001. Statistically significant *P* values are overlaid on the heatmaps. N.D. not determined, N.T. not tested. Blue indicates increased affinity, and red indicates decreased affinity.

## Table 1 | NanoBRET XAC-630 saturation binding parameters (specific binding)

| A$_3$ adenosine receptor | pK$_d$ (n) | P value[a] | $B_{max}$ as% WT | P value[a] |
|---|---|---|---|---|
| WT | 7.2 ± 0.1 (6) | – | 99.3 ± 0.7 | – |
| Y15A[1.35] | 6.8 ± 0.1 (5) | 0.0347 | 28.8 ± 5.7 | <0.0001 |
| S73A[2.65] | 6.9 ± 0.1 (3) | 0.2205 | 85.7 ± 5.8 | 0.8987 |
| T94A[3.36] | 7.0 ± 0.04 (3) | 0.6910 | 37.5 ± 2.4 | 0.0003 |
| H95A[3.37] | 6.6 ± 0.07 (4) | 0.0006 | 34.7 ± 13.4 | <0.0001 |
| H95F[3.37] | 6.4 ± 0.2 (3) | 0.0001 | 34.3 ± 3.0 | 0.0001 |
| V169E[45.53] | 6.9 ± 0.01 (3) | 0.3234 | 83.5 ± 2.2 | 0.8027 |
| M174A[5.35] | 6.7 ± 0.04 (5) | 0.0040 | 54.0 ± 16.5 | 0.0020 |
| N250A[6.55] | N.D. (3) | – | N.D. | – |
| Y265A[7.36] | 7.4 ± 0.03 (3) | 0.8202 | 102.0 ± 6.2 | >0.9999 |
| S271A[7.42] | 7.6 ± 0.1 (3) | 0.0799 | 90.6 ± 1.7 | 0.9925 |
| H272A[7.43] | N.D. (3) | – | N.D. | – |

pK$_d$ values represent the mean ± SEM of the negative logarithm of the equilibrium dissociation constant from (n) experiments performed in duplicate.
*ND* not determined due to no measurable specific binding. $B_{max}$ values were normalised to WT for each experiment.
[a]Significant differences vs WT were determined by one-way ANOVA (Prism 10.3.1) with a Dunnett's multiple comparison post hoc test (*P* < 0.05).

bond interaction with conserved residue N[6.55]. The orientation of the xanthine scaffolds was similar for all the ligands, with DU172 and PSB36 extending deeper into the orthosteric pocket (Fig. 3A; red arrow). There was a slight tilt in the position of the xanthine scaffolds towards TM3 at each receptor subtype relative to the A$_3$AR (Fig. 3A–C; orange arrows). The largest tilt (~42°) was observed with XAC at the A$_{2A}$AR, followed by a 30° tilt for DU172 and PSB36 relative to the A$_3$AR, while the other A$_{2A}$AR xanthines were tilted 15–20°. Given that XAC was used as a fluorescent probe in our NanoBRET experiments, we performed induced fit docking (IFD) with XAC and the A$_3$AR structure (Fig. 3C). The pose of the xanthine closely matched that of LUF7602, with the polar tail extending towards TM1, TM2, and TM7. The difference in the pose of XAC between the A$_3$AR and A$_{2A}$AR could be due to XAC binding in multiple conformations, as the electron density for XAC was not complete in the A$_{2A}$AR structure[33].

DU172 and LUF7602 are modestly selective A$_1$AR and A$_3$AR covalent antagonists with similar chemical structures, binding affinities (DU172: pK$_i$ = 7.4 at A$_1$AR; LUF7602: pK$_i$ = 7.2 at A$_3$AR), and binding modes (Fig. 3D). Both ligands form a covalent linkage with residue Y[7.36] and have a propyl group on N$^3$ that forms hydrophobic contacts with L[3.33], T[3.36], L[6.51], M177[5.38], and W[6.48]. In addition, both ligands have chemical groups that extend off the C$^8$ position into a pocket created by ECL2, TM6, and TM7. In the case of DU172, the larger piperazine forms hydrophobic interactions with E172[45.53], M177[5.35], L253[6.54], N254[6.55],

T257[6.58], and T270[7.35]. This pocket is less conserved among adenosine receptor subtypes, and residue T270[7.35] was shown to contribute to the subtype selectivity of DU172. In contrast, the methoxy group of LUF7602 only interacts with V169[45.53] and L264[7.35]. Residue V169[45.53] is an E at all other adenosine receptor subtypes, which we hypothesised would cause a steric clash with LUF7602 (Fig. 3D). However, the V169E[45.53] mutation did not affect LUF7602 binding (Fig. 2I, J), suggesting the residue adopted a different rotamer.

### Active-state structures of the A$_3$AR
Next, we sought to determine the active-state structures of the A$_3$AR in complex with the G protein. To determine the structure of the A$_3$AR bound to endogenous agonist adenosine, we used the single-chain antibody scFv16[65] and the dominant negative form of Gα$_{i1}$ (DNGα$_{i1}$) to stabilise the complex[66,67]. Single-particle cryo-EM analysis of the purified complex samples yielded a 2.9 Å map providing detailed insights into the A$_3$AR–DNG$_{i1}$–scFv16–adenosine complex (Figs. 4A, B, S5B, and S6 and Table S1).

To obtain a higher-resolution structure with Piclidenoson bound to the A$_3$AR, we used an engineered mini-G$_{si}$ construct[68] that was stabilised using Nb35[65,69] (Fig. S7A). For the A$_3$AR–mG$_{si}$–Nb35–Piclidenoson complex, we achieved a global resolution of 2.5 Å. Following local refinement, the receptor map achieved a resolution of 3.3 Å (Figs. 4C, D and S5C, S7 and Table S1). Both the ECL and ICL regions were well-resolved and displayed nearly identical conformations, allowing clear identification of most receptor residues, except for residues 208–225 in ICL3, which remained disordered.

### Comparison of the active conformation at adenosine receptors
The adenosine- and Piclidenoson-bound A$_3$AR structures closely resemble other active-state adenosine receptor structures with RMSDs of less than 1.0 Å. The TM helices aligned well, with minor divergence at the top of TM7 in the A$_{2A}$AR and A$_{2B}$AR due to a longer ECL3 (Fig. 4E). Despite moderate overall sequence similarity, the pose and position of adenosine in the orthosteric site were highly similar across all adenosine receptor structures (Fig. 4F). The ribose ring extends deeply into the binding site, anchored by a hydrogen bond network involving residues in TM 3, 6, 7 and ECL2. The conserved interaction network between adenosine and ARs includes a π-stacking interaction with F[45.52], a bidentate hydrogen bond with N[6.55], hydrogen bonds between the 3', and 5' hydroxyl groups with S/T[7.42] and H[7.43], and numerous hydrophobic interactions (Fig. 4F, G).

Piclidenson, an A$_3$AR-selective agonist, is a larger molecule than adenosine and extends from the orthosteric site out towards ECL2 and TMs 4–6. Piclidenoson differs from adenosine by two key substituents: a methylcarboxamide group extending from the C5' position and an N$^6$-iodobenzyl group (Fig. 5A–C)[70,71]. The methylcarboxamide group forms a hydrogen bond with T94[3.36], with the methyl group extending into a hydrophobic pocket composed of residues L91[3.33], H95[3.37],

**Table 2 | NanoBRET XAC-630 competition binding parameters**

| A$_3$ adenosine receptor | LUF7602 pK$_i$ (n) | Adenosine pK$_i$ (n) | Piclidenoson pK$_i$ (n) | NECA pK$_i$ (n) |
|---|---|---|---|---|
| WT | 7.5 ± 0.05 (4) | 5.0 ± 0.10 (4) | 7.5 ± 0.08 (5) | 6.3 ± 0.1 (4) |
| Y15A$^{1.35}$ | 6.6 ± 0.2 (4) | 3.9 ± 0.1 (4) | 6.1 ± 0.2 (5) | N.D. (3) |
| S73A$^{2.65}$ | 7.2 ± 0.06 (3) | 4.60 ± 0.1 (3) | 6.6 ± 0.1 (4) | 5.4 ± 0.1 (3) |
| T94A$^{3.36}$ | 7.3 ± 0.05 (3) | 4.9 ± 0.2 (3) | 7.1 ± 0.1 (4) | 4.6 ± 0.1 (3) |
| H95A$^{3.37}$ | 6.2 ± 0.2 (3) | 3.7 ± 0.05 (3) | 5.7 ± 0.3 (3) | 4.0 ± 0.1 (3) |
| H95F$^{3.37}$ | 6.4 ± 0.09 (3) | 3.9 ± 0.2 (3) | 5.7 ± 0.1 (4) | N.D. (4) |
| V169E$^{45.53}$ | 7.2 ± 0.05 (3) | 5.0 ± 0.08 (3) | 8.1 ± 0.2 (4) | 6.00 ± 0.2 (3) |
| M174A$^{5.35}$ | 6.5 ± 0.09 (3) | 4.3 ± 0.5 (3) | 7.9 ± 0.3 (4) | 5.3 ± 0.4 (3) |
| Y265A$^{7.36}$ | 6.3 ± 0.07 (3) | 4.50 ± 0.04 (3) | 6.7 ± 0.1 (4) | 5.0 ± 0.05 (3) |
| S271A$^{7.42}$ | 7.9 ± 0.09 (3) | N.D. (3) | 5.4 ± 0.1 (5) | N.D. (3) |

Values represent the mean ± SEM from (n) experiments performed in duplicate.
*ND* not determined due to no measurable specific binding.

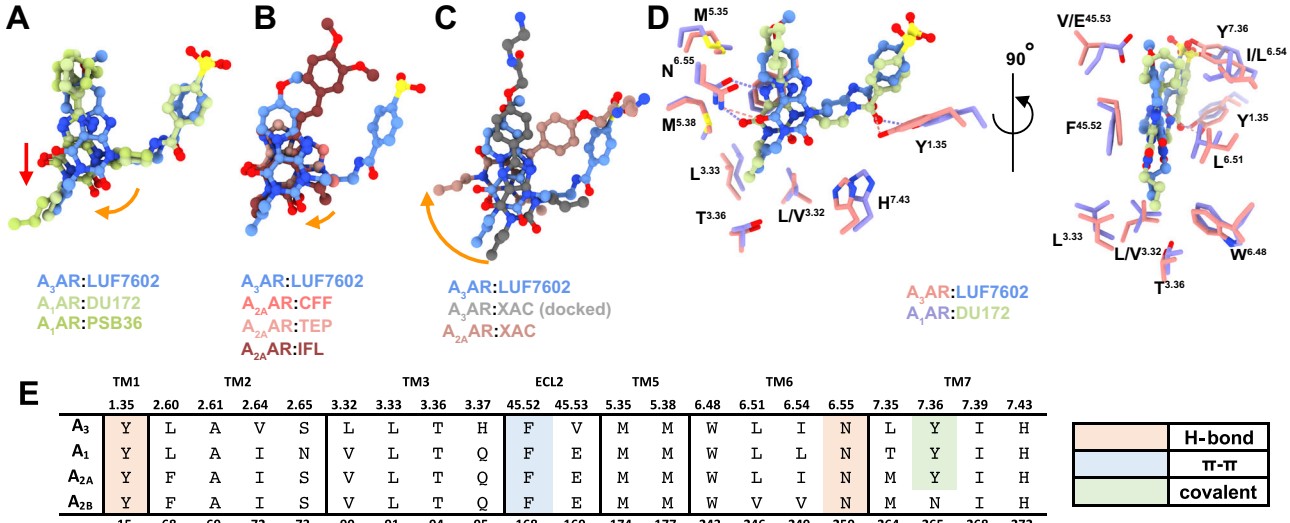

**Fig. 3 | Comparison of the binding pose of adenosine receptor antagonists.**
A–C Binding poses of ligands from different antagonist-bound adenosine receptor structures. The red arrow indicates a displacement, and the orange arrows indicate rotations. A Comparison of A$_3$AR:LUF7602 with A$_1$AR:DU172 (PDB: 5UEN, light green) and A$_1$AR:PSB36 (PDB: 5N2S, light green). B Comparison of A$_3$AR:LUF7602 with A$_{2A}$AR:CFF (PDB: 5MZP, peach), A$_{2A}$AR:TEP (PDB: 5MZJ, pink), and A$_{2A}$AR:IFL (PDB: 8GNG, red). C Comparison of A$_3$AR:LUF7602 with A$_3$AR:XAC (docked) and A$_{2A}$AR:XAC (PDB: 3REY, dark pink). D Detailed view of ligand-receptor interactions for A$_3$AR:LUF7602 and A$_1$AR:DU172, showing key residues involved in binding. E Sequence alignment of residues involved in receptor interactions from (D).

S181$^{5.42}$, I186$^{5.47}$, and W243$^{6.48}$ (Fig. 5D). Non-conserved residues H95$^{3.37}$ and S181$^{5.4}$ restrict the size of the orthosteric pocket. The importance of this pocket is underscored by the 30-fold decrease in 2-Cl-Piclidenoson (i.e. Namodenoson) binding at the H95$^{3.37}$ alanine mutation[72]. The pocket's proximity to the rotamer toggle switch residue W243$^{6.48}$ suggests a role in receptor activation. The N$^6$-iodobenzyl group occupies a hydrophobic pocket formed by TMs 5–7 and ECL2, making various hydrophobic interactions (Fig. 5E). Notably, the iodine atom points towards the backbone carbonyl of M172$^{45.56}$ in an orientation ($\theta_1$ = 121°, $\theta_2$ = 96°) and distance (3.4 Å) that favours a halogen bond interaction[73]. Structure-activity relationships support this halogen bond, as the replacement of the iodine atom with H or Cl decreases binding affinity by 16- and 20-fold, respectively[71].

To validate the observed agonist binding modes, we measured the binding affinity of adenosine and Piclidenoson using a competition NanoBRET binding assay with XAC-630. We also tested NECA because it is chemically similar to adenosine and Piclidenoson with an ethyl-carboxamido group that extends from the C5' position (Fig. 6A–C and Table 2). The agonists had a binding affinity rank order of Piclidenoson > NECA > adenosine, consistent with previous studies and with

Piclidenoson's more extensive interactions with the receptor compared to adenosine (Fig. 6D, E). Residue S271A$^{7.42}$ caused the largest loss of binding across all three agonists due to the loss of interaction with the 3' OH group. Similarly, H95$^{3.37}$ and H95F$^{3.37}$ significantly affected the binding of all three agonists, particularly NECA, suggesting these mutations impact the orthosteric site. Interestingly, we observed that Y15A$^{1.35}$ significantly affected the binding for all three agonists despite not forming any direct interactions. This residue forms part of a hydrogen bond network between TM1, TM2, and TM7 that helps coordinate H272$^{7.43}$ with the ribose group (Fig. 7E). Mutations S73A$^{2.65}$ and Y265A$^{7.36}$ had similar effects, though less apparent with adenosine. Residue Y265$^{7.36}$ contributes to the hydrogen bond network by forming a π-stacking interaction with Y15$^{1.35}$. The role of S73$^{2.65}$ was more subtle, but it can hydrogen bond to Y265$^{7.36}$ when adopting a different rotamer (Fig. 7E).

In contrast, the T94A$^{3.36}$ mutation only significantly reduced the binding of NECA, suggesting it forms a key interaction with the ethylcarboxamido group (Fig. 6F). Similar to LUF7602, the ECL2 V169E$^{45.53}$ mutation was designed to also test the selectivity of Piclidenson. However, the mutation resulted in a 5-fold increase in binding

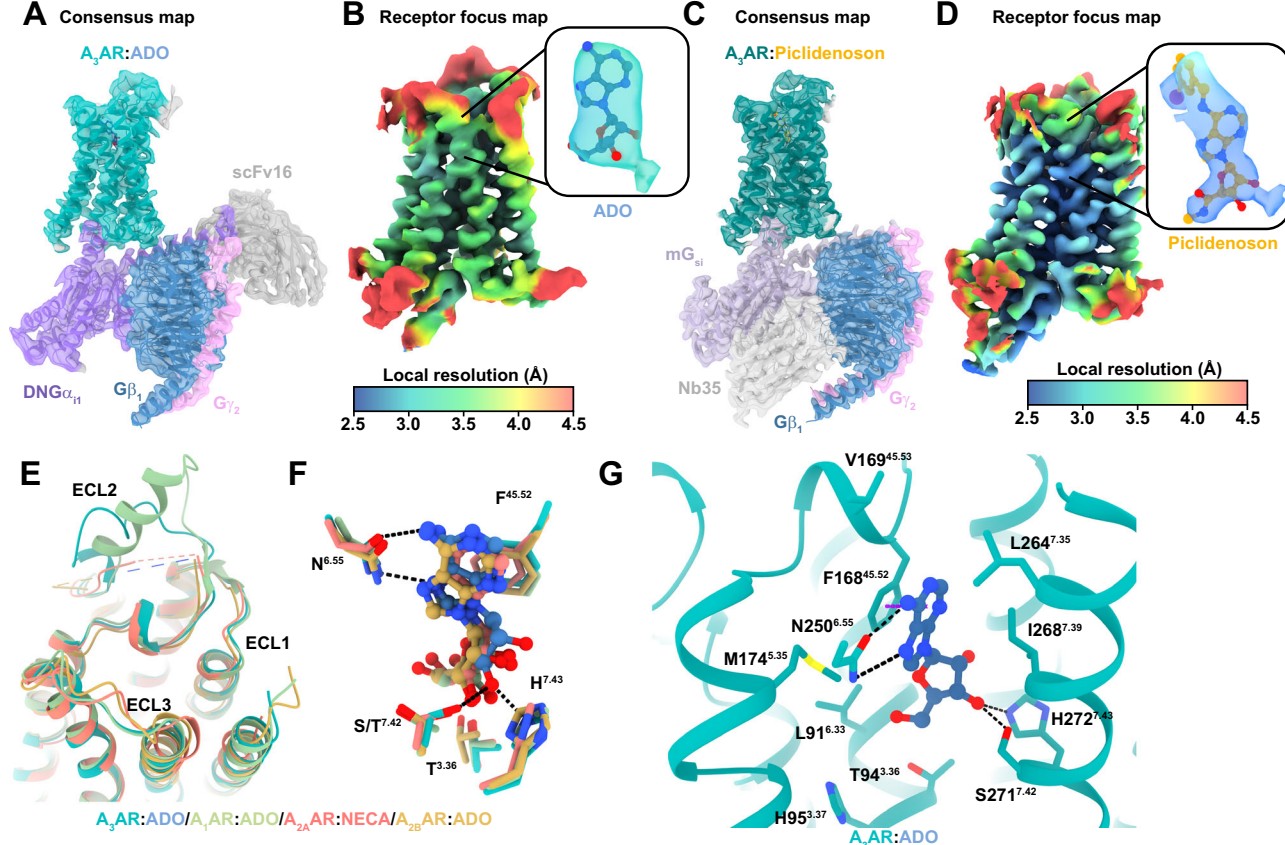

**Fig. 4 | Comparison of A3AR cryo-EM structures in the active conformation.**
**A** Consensus cryo-EM density map (contour = 0.35) and atomic model of the active A3AR:adenosine (ADO) complex, showing the receptor (green), ADO (blue), DNGαi1 (purple), Gβ1 (dark blue), Gγ2 (pink), and scFv16 (grey). **B** Local resolution receptor-focused cryo-EM map (contour = 0.3) with an inset of the density around ADO. **C** Consensus cryo-EM density map (contour = 0.35) and atomic model of the active A3AR:Piclidenoson complex, showing the receptor (dark green), Piclidenoson (orange), mGsi (light purple), Gβ1 (dark blue), Gγ2 (pink), and Nb35 (grey).

**D** Local resolution receptor-focused cryo-EM map (contour = 0.35) with an inset of the density around Piclidenoson. **E** Extracellular view of aligned structures of A3AR:ADO (cyan), A1AR:ADO (PDB: 7LD4, green), A2AAR:NECA (PDB: 6GDG, red), and A2BAR:ADO (PDB: 8HDP, orange). **F** Comparison of adenosine binding in A3AR:ADO (cyan), A1AR:ADO (PDB: 7LD4, green), A2AAR:ADO (PDB: 2YDO, red), and A2BAR:ADO (PDB: 8HDP, orange), highlighting key interacting residues.
**G** Detailed view of A3AR:ADO interactions, showing key residues involved in ligand binding.

affinity for Piclidenoson (Fig. 6F). Similar results for the V169E[45.53] mutant were observed in a prior study that included the examination of the V169E[45.53] mutant in molecular dynamics simulations[74]. The simulations suggested an increase in hydrophobic interactions with M174[5.35], M177[5.38], and I253[6.58]. Another possible explanation for the increase in affinity could be the creation of an anion-aromatic interaction[75] between E169 and the edge of the N[6]-iodobenzene group.

The mutation of residue M174A[5.35] had paradoxical effects. With adenosine and NECA, M174A[5.35] had a slightly reduced affinity and showed non-competitive displacement (Fig. 6A, B). For Piclidenoson, M174A[5.35] increased affinity and enhanced displacement of XAC-630 (Fig. 6C). These results were consistent with a prior study[74]. These results also align with our cryo-EM data, which shows that the conformation of M174[5.35] is influenced by the specific ligand occupying the orthosteric binding site (Fig. 5F). In the Piclidenoson-bound structure, M174[5.35] is pushed back towards TM6 by ~2–3 Å compared to adenosine- and LUF7602-bound structures and to M[5.35] in structures of the A1AR, A2AAR, and A2BAR. Thus, M174[5.35] may function as a gatekeeper residue, controlling access to a cryptic extracellular pocket. This cryptic pocket may also facilitate the non-specific binding of XAC-630, as the position of the 2-aminoethyl-acetamide group of XAC appeared flexible when we docked XAC-630 into our inactive A3AR structure, potentially explaining the observed non-competitive inhibition with the M174A[5.35] mutation. Recently published structures of Piclidenoson-bound A3AR structure (PDB: 8X16)[37] and Namodenoson-bound to the

sheep A3AR (PDB: 8YH6)[38] reveal the N[6]-iodobenzene group modelled oriented towards the solvent (Figs. S8 and S9), with no corresponding changes to the conformation of M174[5.35]. However, poor map density in the ECL region of 8X16 and 8YH6 suggests potential uncertainties with side chain and ligand modelling.

### Activation mechanism
The comparison of the inactive, LUF7602-bound, and active, agonist-bound A3AR complex structures offers insight into the mechanism of A3AR activation (Fig. 7A–C). Adenosine and Piclidenoson bind deeper in the orthosteric site than LUF7602, triggering conformational changes characteristic of Class A GPCRs[76]. These changes involve a 2.5 Å downward shift in the rotamer toggle switch residue W243[6.48], leading to signal propagation through the PIF, NPxxY, and DRY motifs, followed by an ~11 Å outward movement of TM6 that is typical of Class A Gi coupled receptors[77]. While LUF7602, adenosine, and Piclidenoson interact similarly with orthosteric site residues (F168[45.42], N250[6.55], and H272[7.43]), their binding modes and resulting A3AR conformations differ significantly. In the active state, TM1, TM2, ECL1, and TM7 move inward at the extracellular side (Fig. 7B), breaking the ECL2 anti-parallel β-sheet observed in the inactive conformation.

A key feature of the active conformation is an extensive hydrogen bond network that extends from the top of TM7 through TM1, TM2, and down to the ribose binding site (Fig. 7D, E). Specifically, Q261[7.31] and N12[1.32] form hydrogen bonds with Y265[7.36], positioning it to form a

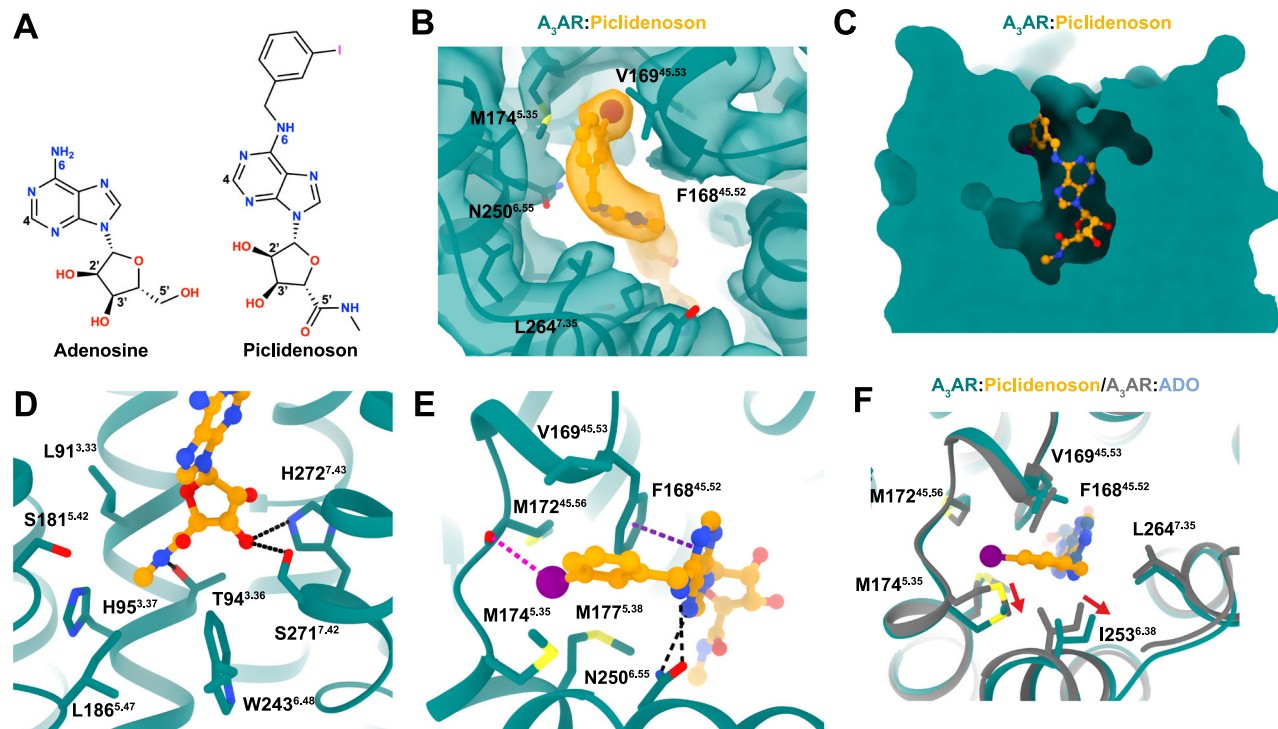

**Fig. 5 | The Piclidenoson binding site. A** Chemical structures of adenosine and Piclidenoson with key atoms numbered. **B** Cryo-EM density of receptor refined map (contour = 0.35) of $A_3AR$ Piclidenoson-bound binding site. **C** Surface representation of the $A_3AR$ orthosteric binding pocket with Piclidenoson shown as orange sticks. **D, E** Detailed view of $A_3AR$:Piclidenoson interactions, showing key residues involved in ligand binding near (**D**) the ribose group and **E** the $N^6$-iodobenzyl group. **F** Comparison of the position of residue $M174^{5.35}$ in the adenosine-bound (coloured grey) and Piclidenoson-bound structures.

π–π stacking interaction with $Y15^{1.35}$. Residue $Y15^{1.35}$ forms a hydrogen bond with $E19^{1.39}$, which forms a hydrogen bond with $H272^{7.43}$ that positions $H272^{7.43}$ for interaction with the 2′ and 3′ OH groups of adenosine and Piclidenoson. The 3′OH group of the agonists also interact with $S271^{7.42}$, while the 5′ group may interact with $T94^{3.3.6}$ (Fig. 7E). We note that disruption of this hydrogen bond network by mutation of $Y15^{1.35}$, $S73^{2.65}$, $S271^{7.42}$, and $Y265^{7.36}$ significantly reduced agonist binding (Fig. 6F). Notably, this extensive hydrogen bond network is absent in the LUF7602-bound structure, partly due to LUF7602's covalent interaction with $Y265^{7.36}$ disrupting the conformation of nearby residues (Fig. 7D). Consequently, LUF7602 forms fewer interactions with the receptor compared to adenosine and Piclidenoson (Figs. 2F vs 6D, E). The uniqueness of this extensive hydrogen bond network to $A_3AR$, not observed in other AR subtypes, suggests a distinct activation mechanism for this receptor subtype.

To test the role of residues surrounding the agonist-binding site on receptor activation, we used the BRET-based Trupath assay[78] to measure the proximal activation of $G\alpha_{i1}$ (Fig. 7F and Table S2). The efficacy of the agonists ($\tau_A$) was calculated using the Black–Leff operational model of agonism[79] and corrected for differences in receptor expression[80] (Fig. 7G and Table S3). The agonists had a rank order of efficacy with adenosine > NECA > Piclidenoson. Similar to our findings from binding experiments, the $N250A^{6.55}$ and $H272A^{7.43}$ mutants did not signal. Similarly, NECA and Piclidenoson did not activate $S271A^{7.42}$, while adenosine produced a weak response that could not be quantified due to the absence of measurable binding affinity (Fig. 6A). In addition, $H95F^{3.37}$ did not produce a measurable response, suggesting that although agonists could bind to this receptor mutant, they could not activate the receptor. The $H95A^{3.37}$ mutation, however, displayed ligand-dependent effects, producing no response for Piclidenoson, a small response for adenosine, and a robust response for NECA despite a 100-fold loss in binding affinity. The increase in efficacy for NECA at $H95A^{3.37}$ is likely related to its

juxtaposition to the ethylcarboxamido group and the rotamer toggle switch residue $W243^{6.48}$. Similarly, there was a ligand-dependent increase in the efficacy of NECA at the $Y265A^{7.36}$, suggesting that $Y265^{7.36}$ reduces agonist binding but does not alter signalling. Nearby mutation $Y15A^{1.35}$ reduced adenosine efficacy to a similar level as Piclidenoson, indicating a small effect on signalling. Finally, the $M174A^{5.35}$ mutation significantly reduced agonist efficacy to the same levels (Table S3), indicating that $M174^{5.35}$ is important for the binding and signalling of agonists but with less of an effect on Piclidenoson. Overall, these orthosteric site mutations reveal distinct roles for binding site residues in receptor activation, with some mutations completely abolishing signalling while others show ligand-dependent effects on efficacy.

## G protein interface

Adenosine receptor subtypes exhibit different G protein coupling preferences[81], with $A_1AR$ and $A_3AR$ preferentially coupling to $G_i$ proteins and $A_{2A/2B}AR$ coupling to $G_s$ proteins (Fig. 8A). Our study employed two different G protein constructs: a dominant negative $G_{i1}$ for the adenosine-bound structure and a mini-$G_{si}$ chimera for the Piclidenoson-bound structure (Fig. S7A). Comparing the adenosine- and Piclidenoson-bound structures revealed a similar set of interactions with the last five residues of the G protein α5-helix and the $A_3AR$ core (Fig. 8B–D). Beyond these five residues, the interactions between G proteins and $A_3AR$ diverge, with the α5-helix of $G_{i1}$ rotating away by 5 Å near the end of the α5-helix (Fig. 8E). Globally, this can be viewed as a rotation of the G proteins around the core of the receptor, resulting in a large displacement between the αN-helices (Fig. 8F). On the one hand, these differences may be due to the different G protein constructs that were used. For example, the conformation of the α5-helix in the adenosine-$A_3AR$-$G_{i1}$ structure was more similar to the $A_1AR$-$G_{i1}$ structure and a recent Piclidenoson-$A_3AR$-$G_i$ structure (PDB: 8X16), while the α5-helix in our Piclidenoson-$A_3AR$-mini-$G_{si}$ structure was

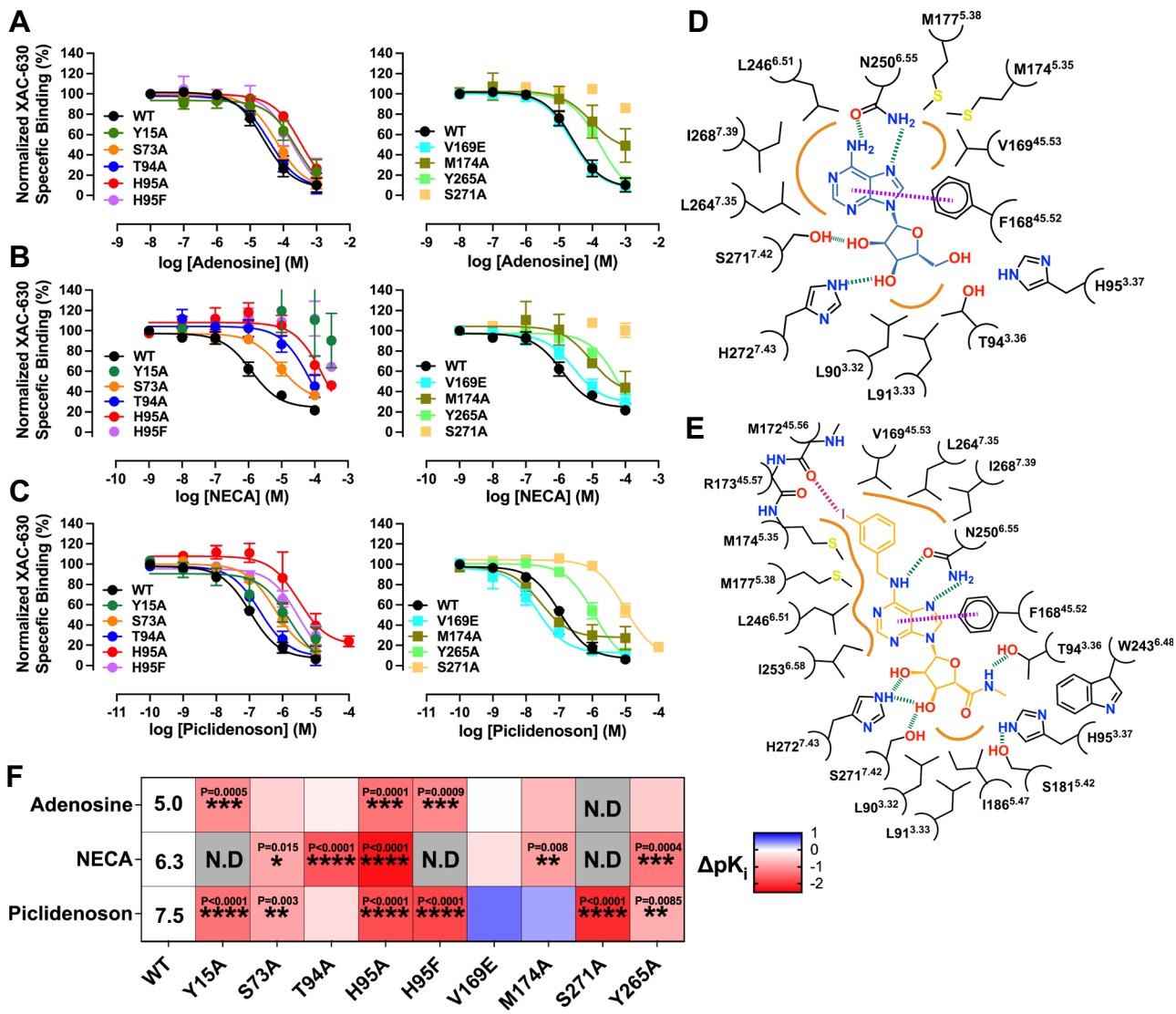

**Fig. 6 | Key residues in the agonist-binding site. A–C** Competitive binding curves showing the displacement of XAC-630 by (**A**) adenosine, **B** NECA, and **C** Piclidenoson at WT and mutant A₃AR constructs. Grouped data are shown as mean ± SEM, with experiments performed in duplicate. Grouped pKᵢ values are shown in Table 2. **D, E** 2D interaction diagram of **D** the A₃AR:ADO complex and **E** the A₃AR:Piclidenoson complex. Hydrogen bonds are shown as green dashed lines, a π–π stacking interaction as purple dashed lines, a halogen bond as a pink dashed line, and orange lines as hydrophobic interactions. Residue numbering follows the Ballesteros-Weinstein convention. **F** Heat map showing the effects of various A₃AR mutations on binding affinity (ΔpKᵢ) for adenosine, NECA, and Piclidenoson. The

pKᵢ for adenosine, NECA, and Piclidenoson at WT is overlaid on the heatmap. Statistical analysis was from $n = 3$ for all except for WT ($n = 4$) and Y15A ($n = 4$) with adenosine, WT ($n = 4$) and H95F ($n = 4$) with NECA, and $n = 4$ (S73A, T94A, H95F, V169E, M174A, Y265A), $n = 5$ (WT, Y15A, and S271A) with Piclidenoson. Significant differences vs WT were determined by one-way ANOVA with a Dunnett's multiple comparison post hoc test. P values: * = 0.01–0.05; ** = 0.001–0.01; *** = 0.0001–0.001; **** = <0.0001. Statistically significant P values are overlaid on the heatmaps. (F). N.D. not determined. Blue indicates increased affinity, and red indicates decreased affinity.

more similar to the A₂ₐ/₂ᵦAR-Gₛ structures. On the other hand, the conformation of the αN-helix in the adenosine-A₃AR-Gᵢ₁ appears to be an outlier compared to the other adenosine receptor structures, suggesting this could be specific to adenosine-A₃AR-Gᵢ₁. Despite the two Piclidenoson-bound structures using different G protein constructs and stabilising methods, Nb35 vs LgBit-HiBit tethering[82], a similar number of contacts were made between the A₃AR and G protein (Fig. 8C). Notably, the adenosine-bound structure shows fewer receptor-G protein contacts compared to the Piclidenoson-bound structure, potentially revealing ligand-dependent A₃AR-G protein conformations. We caveat these statements by highlighting the differences in G protein constructs and stabilising methodologies that are commonly used in cryo-EM studies, and further studies will be required.

## Discussion

The human A₃AR has attracted considerable interest as a drug target for treating various indications, as evidenced by multiple drugs now progressing through clinical trials. Given the clinical potential of the A₃AR, significant effort was invested into identifying potent and selective A₃AR agonists, antagonists, and allosteric modulators[17,83–86]. To understand how potential drugs bind and modulate the activity of the A₃AR, we determined cryo-EM structures of the A₃AR in the inactive and active conformations.

The structure of A₃AR bound to the covalent antagonist LUF7602 revealed a binding mode that shares similarities with other xanthine-based antagonists bound to A₁AR and A₂ₐAR, including the A₁AR covalent antagonist DU172[32,33]. The covalent attachment to Y265⁷·³⁶, while crucial for structure determination, appears to disrupt optimal

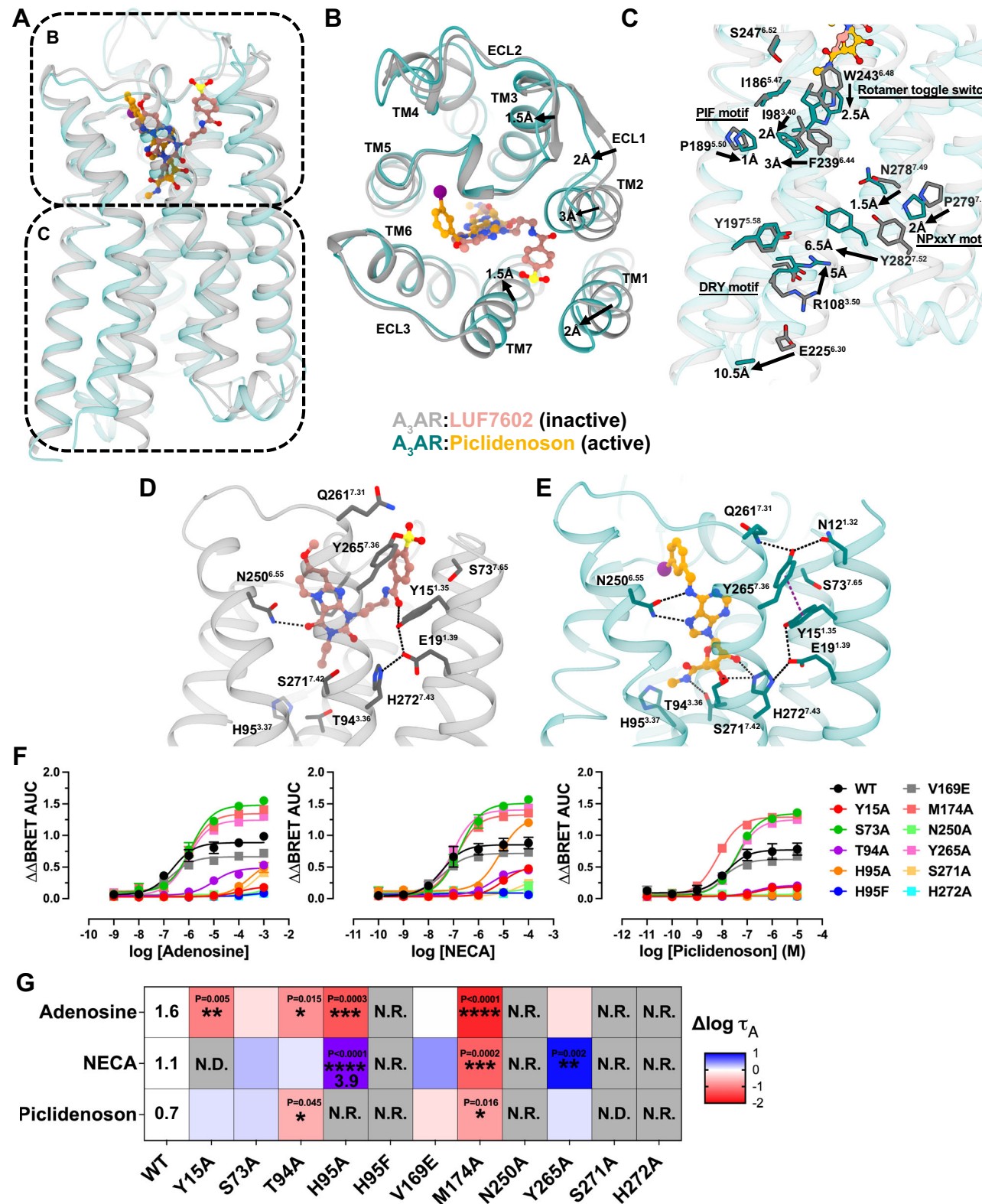

interactions with key residues N250[6.55] and H272[7.43]. Although the covalent properties of DU172 and LUF7602 were essential to structure determination, both ligands have lower binding affinities than their parent molecules, DPCPX and a pyrido-[2,1-f]purine-2,4-dione[49,87]. This suggests that the covalent attachment of the ligands may result in less favourable interactions with the receptor. Future structures or computational studies with the parent molecules could provide further insight and aid in the discovery of better antagonists.

We were not successful in determining a high-resolution cryo-EM structure of the A₃AR with the selective antagonist MRS1220. Structural studies focusing on selective A₃AR antagonists remain crucial for determining the mechanisms underlying selective A₃AR inhibition. The inability to determine a high-resolution MRS1220-bound A₃AR structure was potentially a limitation of using the BRIL fusion approach[40], where an antibody fragment and Nb provide stability to the complex. Though these regions align well, there is a loss in resolution moving

**Fig. 7 | Activation mechanism of the A₃AR. A** Overall structure of A₃AR in active (green) and inactive (light blue) states. The top panel shows the extracellular view in (**B**), and the bottom panel shows the side view in (**C**). **B** Detailed comparison of active and inactive A₃AR structures, highlighting key conformational changes viewed from the extracellular surface. Arrows indicate the direction and magnitude of movements. **C** Close-up view of the intracellular region, showing key residues and motifs involved in receptor activation. Signalling motifs such as the PIF motif, rotamer toggle switch, NPxxY motif, and DRY motif are labelled. Arrows indicate conformational changes with distances. **D, E** The binding pocket of **D** the inactive A₃AR bound to LUF7602 and **E** the active A₃AR bound to Piclidenoson. Key residues involved in ligand interactions are shown and labelled. Hydrogen bonds are depicted as black dashes and π–π stacking interactions as purple dashed lines. **F** Gα$_{i1}$ activation using the TruPath assay. Trupath experimental data for each agonist was baseline-corrected to the initial baseline and then to the buffer control

($\Delta\Delta$) followed by calculating the area under the curve (AUC) for each response ($\Delta\Delta$BRET AUC). Data points represent grouped mean ± SEM values. A three-parameter log[agonist] vs response model was fit to the data. Data are from $n = 3$ experiments performed in duplicate. **G** Heat map showing the effects of various A₃AR mutations on signalling efficacy ($\Delta$log $\tau_A$) for adenosine, NECA, and Piclidenoson. Log $\tau_A$ for adenosine, NECA, and Piclidenoson at WT and NECA at H95A are overlaid on the heatmap. Significant differences vs WT were determined by one-way ANOVA with a Dunnett's multiple comparison test. $P$ values: * = 0.01–0.05; ** = 0.001–0.01; *** = 0.0001–0.001; **** = <0.0001. Statistically significant $P$ values are overlaid on the heatmaps (**G**). N.D. = $\tau_A$ was not determined due to the lack of a K$_A$ value. N.R. no response. Blue indicates increased efficacy, red indicates decreased efficacy, and the purple square for NECA indicates a much higher log $\tau_A$ value. Values are reported in Table S3.

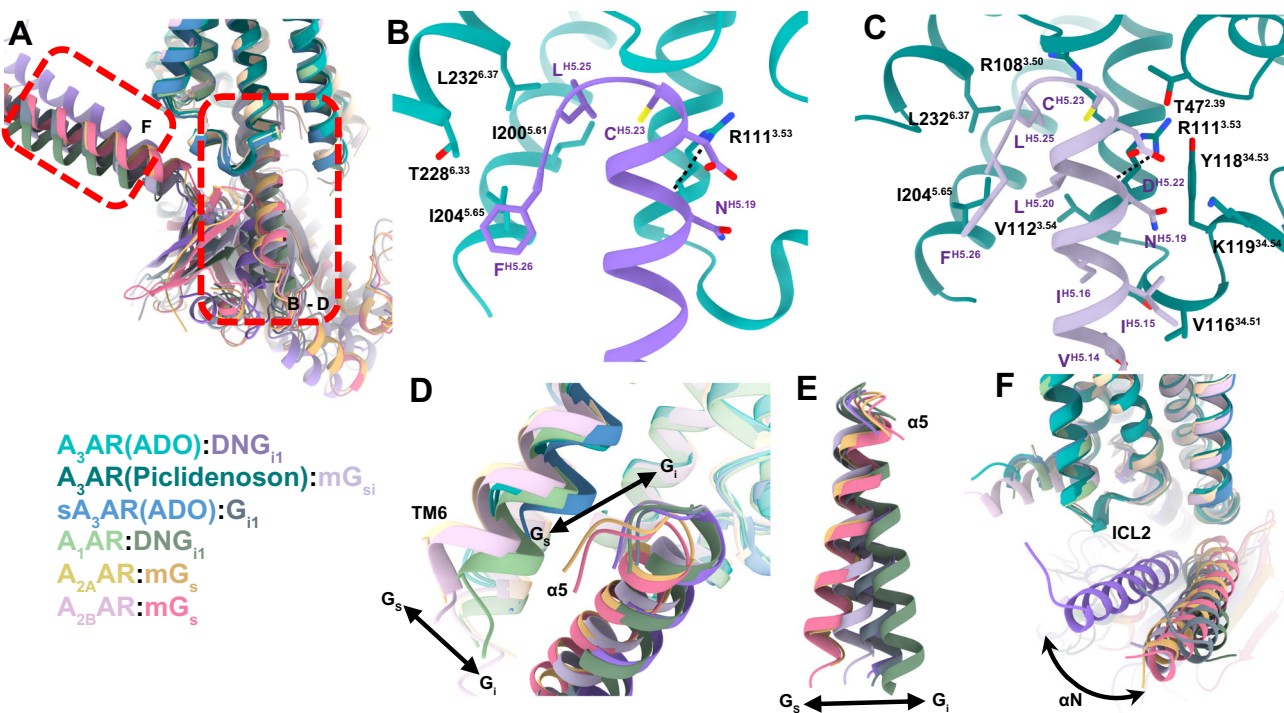

**Fig. 8 | Comparison of the adenosine receptor G protein interface.**
**A** Comparison of adenosine receptor−Gα complexes. The view is of the interaction of Gα with the receptor. Proteins are coloured according to the legend. The red dashed box indicates regions shown in panels B-F. Structures shown and coloured as: A₃AR:ADO:DNG$_{i1}$ (coloured cyan and purple, PDB: 9EBH); A₃AR:Piclidenoson:mG$_{si}$ (coloured green and light purple, PDB: 9EBI); sheep A₃AR:ADO:G$_{i1}$ (coloured blue and grey, PDB: 8YH2); A₁AR:ADO:DNG$_{i1}$ (coloured light green and dark green, PDB: 7LD4); A$_{2A}$AR:NECA:mG$_s$ (coloured yellow and orange, PDB: 6GDG); and A$_{2B}$AR:ADO:mGs (coloured light pink and red, PDB: 8HDP, orange). **B, C** Detailed view of the A₃AR:G protein interface from **B** the adenosine-

bound A₃AR-DNG$_{i1}$ complex and **C** the Piclidenoson-bound A₃AR-mGsi complex. Receptor residues are shown in shades of green, and G protein residues are in shades of purple. Receptor residues are numbered according to Ballesteros and Weinstein numbering, and G proteins are numbered according to the CGN scheme. **D** Comparison of G protein coupling in different adenosine receptor structures, highlighting the positions of TM6 and the α5 helix. **E** Overall Comparison of the α5 helix orientation in different G$_s$ and G$_i$ protein-bound adenosine receptor structures. **F** Comparison of the αN helix from the adenosine-bound A₃AR structure (purple) with other adenosine receptor:G protein complexes.

towards the more dynamic regions of the A₃AR, which include the orthosteric site and the ECL region. In contrast, the addition of the covalent antagonist LUF7602 provided sufficient stability for structure determination, albeit the overall resolution surrounding the ligand binding site was at moderate resolution. Other methods to stabilise the inactive state of the receptor by providing a fiducial marker, such as Nb6[88] or fusions with the C-terminus[89], may provide alternative approaches for future efforts.

The structures of A₃AR bound to the endogenous agonist adenosine and the clinically relevant agonists Namodenoson and Piclidenoson provide valuable insights into the molecular basis of agonist recognition and selectivity[37,38]. The binding mode of adenosine is highly conserved across AR subtypes, consistent with its role as the

endogenous ligand. During the preparation of our manuscript, a cryo-EM structure of the sheep A₃AR bound to adenosine by Oshima et al.[38] revealed a similar binding pose and interactions with the receptor (Fig. S10D), further confirming this conservation.

Subtype-selective A₃AR agonists have been designed by introducing modifications at the N$^6$ and C$^2$ positions of adenosine. Modifications at the C$^2$ position, such as the Cl group in Namodenoson, extend towards TM2 (Fig. S10C). Similarly, the A₃AR selective antagonist LUF7602 and the A₁AR selective antagonist DU172 both contain moieties that extend towards TM2. Our comparison of inactive and active A₃AR structures revealed an outward movement of TM2 at the extracellular side of the receptor. This conformational change appears to be associated with ligand binding rather than receptor activation, as

evidenced by the convergence of TM2 conformations near D58$^{2.50}$ in both states. Supporting this interpretation, molecular modelling studies with bulkier C$^2$-extended agonists[90,91] predicted similar TM2 movements. These observations suggest that TM2 flexibility accommodates diverse C$^2$ substituents and contributes to ligand selectivity without directly participating in receptor activation mechanisms.

Both Namodenoson and Piclidenoson contain an N$^6$-iodobenzyl group critical for A$_3$AR selectivity[92]. Previous studies indicated that this N$^6$-substituent binds to a hydrophobic pocket formed by TM5, TM6, and ECL2[74]. Interestingly, in recent cryo-EM structures of A$_3$AR complexed with these agonists, the position of the N$^6$-iodobenzyl group was poorly resolved due to limited density. Our Piclidenoson-bound structure revealed that this N$^6$-iodobenzyl group occupies what can be described as a cryptic pocket, a binding site that only becomes accessible through conformational changes, in this case through the movement of the conserved gatekeeper residue M174$^{5.35}$. Such cryptic pockets are rarely captured in experimental structures and are typically identified through molecular dynamics simulations[93–97]. The discovery of this cryptic pocket accommodating the N$^6$-iodobenzyl group not only explains the structural basis for A$_3$AR selectivity but also provides an exciting opportunity for the rational design of more selective A$_3$AR agonists. This finding underscores the value of determining multiple structures of the same protein-ligand complex, as it can reveal dynamic features and conformational heterogeneity that might be missed in a single structure. Collectively, our results, along with other recent cryo-EM studies, indicate that A$_3$AR ligands can achieve subtype selectivity through two distinct mechanisms: C$^2$ modifications that interact with the TM2 region and N$^6$ substitutions that access a cryptic pocket between TM5, TM6, and ECL2. The study by Cai et al. further suggests that ECL3 may contribute to selectivity, potentially due to its proximity to both binding pockets[37].

Comparison of our inactive and active A$_3$AR structures revealed a unique activation mechanism characterised by an extensive hydrogen bond network extending from the top of TM7 through TM1, TM2, and down to the ribose binding site. This network, not observed in other AR subtypes, appears to play a role in facilitating the conformational changes associated with A$_3$AR activation. With regards to receptor activation, multiple mutations impaired receptor signalling, while other mutations showed ligand-dependent effects on efficacy, particularly with the agonist NECA. Further exploration of the unique A$_3$AR activation mechanism, possibly through additional mutagenesis studies, signalling assays, and molecular dynamics simulations, is warranted[98–101]. In addition, structural studies of the A$_3$AR bound to a wider range of ligands, including C2 extended agonists, partial agonists, and allosteric modulators, will further elucidate the structural basis of ligand efficacy, selectivity, and allosteric modulation[30,102]. Together, these insights could be valuable for the design of novel ligands with tailored efficacy profiles[103,104].

Our structures of A$_3$AR in complex with different G protein constructs revealed both conserved and divergent features of the receptor-G protein interface. The core interactions involving the last five residues of the G protein α5-helix appear to be consistent across different ligands and G protein constructs. However, the observed differences in the orientation of the α5-helix and αN-helix between the adenosine- and Piclidenoson-bound structures raise intriguing questions about ligand-dependent modulation of G protein coupling. The reduced number of contacts between A$_3$AR and G protein in the adenosine-bound structure compared to the Piclidenoson-bound structure suggests a potential link between the extent of ligand-receptor interactions in the orthosteric site and the stability of the receptor-G protein complex[102]. This observation aligns with the concept of ligand-dependent signalling bias, and the observed differences in G protein coupling between adenosine and Piclidenoson suggest the potential for developing ligands with biased signalling properties[91,105,106]. This could be particularly valuable in therapeutic contexts where selective activation of specific signalling pathways is desired[107–109]. The use of different G protein constructs (DNG$_i$, mG$_{si}$), stabilising methods (scFv16, Nb35), and tethering strategies (C-terminal Gα fusion, LgBit-HiBit tethering between the C-terminus and Gα protein, and Gγ$_2$-G$_{i1}$ fusion) is common in GPCR cryo-EM studies[65,67,82,110–112], but necessitates caution in interpretation and is a general limitation of these studies. Investigation of ligand-dependent G protein coupling using consistent G protein constructs and methodologies to clarify the observed differences between adenosine- and Piclidenoson-bound structures, along with higher resolution cryo-EM maps, will be important for future studies.

This comprehensive study of A$_3$AR structure and pharmacology provides a solid foundation for future research and highlights areas that warrant further research. A key challenge in A$_3$AR drug development was significant variation in ligand pharmacology across different species of receptors[86]. The high-resolution structures presented here offer a structural framework for determining the molecular basis of species-dependent differences in A$_3$AR pharmacology, which would aid future translation studies. In addition, these structures provide an opportunity to reinterpret existing medicinal chemistry and structure-activity relationship data for A$_3$AR ligands[86,90,113,114]. Integration of these structural insights with previous findings will allow for more rational and targeted approaches to ligand design. These advances open new avenues for developing highly selective and efficacious A$_3$AR ligands with potential therapeutic applications across a spectrum of disorders, including inflammatory and liver diseases and glaucoma.

## Methods

### Materials
FreeStyle 293-F suspension cells, FreeStyle 293 expression serum-free media, Dulbecco's modified Eagle's medium (DMEM), foetal bovine serum (FBS), and trypsin were purchased from Invitrogen. Rolipram and forskolin were purchased from Sigma. Guanosine 5′-(γ-thio) triphosphate, [35S]-, 1250 Ci/mmol was purchased from PerkinElmer Life. N$^6$-(3-iodobenzyl)−5′-N-methylcarboxamidoadenosine (IB-MECA/Piclidenoson), 2-Cl-$IB$-$MECA$ was purchased from MedChem Express (Monmouth Junction, USA). NECA (adenosine-5′-N-ethyluronamide), MRS1220 (9-Chloro-2-(2-furanyl)-5-((phenylacetyl)amino)-[1,2,4]triazolo[1,5-c]quinazoline) and adenosine were purchased from Sigma (St. Louis, MO, USA). Adenosine deaminase (ADA) was purchased from Roche Applied Sciences (Mannheim, Germany). Furimazine was purchased from Promega (Alexandria, Australia) and coelentrazine-h from Nanolight Technology (Arizona, USA). Tested compounds were dissolved in dimethyl sulfoxide. Fluorescent conjugates of xanthine amine congener (XAC) CA200645 were purchased from CellAura Technologies Ltd. (Nottingham, UK). For western blots, mouse poly-His antibody was purchased from QIAGEN, and IRDye 680RD goat anti-Mouse IgG was from LI-COR. Lauryl maltose neopentyl glycol (LMNG) and cholesteryl hemisuccinate (CHS) were purchased from Anatrace. Benzonase was purchased from Merck Millipore, and apyrase was from NEB.

### Cell lines
FreeStyle 293-F suspension cells for transient transfection of WT-A$_3$AR, the A$_3$AR expression, and A$_3$AR 3 C construct were grown in FreeStyle 293 expression serum-free media (Gibco) and maintained in a 37 °C incubator containing a humidified atmosphere with 5% CO2 on a shaker platform rotating at 120 rpm. Human HEK 293 adherent (HEK293a) cells for transient transfection of Nluc-A$_3$AR were grown in DMEM with 10% ($v/v$) FBS and maintained at 37 °C with 5% CO$_2$. Chinese hamster ovary (CHO) cells stably expressing the human adenosine A$_3$ receptor (Flp-In-CHO cells) were generated as previously described in ref. 91. Cells were grown in the culture DMEM containing 10% ($v/v$) FBS and selection antibiotic hygromycin B (0.5 mg/mL) at 37 °C with 5% CO2. CHO cells were regularly tested to ensure they were free from

mycoplasma. Insect cells *Spodoptera frugiperda* (Sf9) and *Trichoplusia ni* (TNI) were grown in ESF 921 serum-free media (Expression System) and shaken at 27 °C. Sf9 and TNI cells were not tested for mycoplasma.

## Constructs

The $A_3AR$-BRIL-S97R construct was designed in a pFastBac vector as discussed in the results section. The anti-BRIL Fab (BAG2) and anti-BAG2 Nb were gifts from Christopher Garcia's laboratory[40,115]. The anti-BAG2 Nb was cloned in the pET26b+ vector with an N-terminal histidine tag followed by a TEV protease site. For the $A_3AR$ G protein-bound complexes, the human WT $A_3AR$ (Uniprot ID: P0DMS8) was modified with a FLAG epitope at the N-terminus and a Histidine tag at the C-terminus to facilitate purification and detection was cloned into a pFastBac vector. To improve receptor expression, the first 22 amino acids of the human $M_4$ muscarinic receptor, which contains 3 N-glycosylation sites[32], were inserted between the FLAG tag and the native $A_3AR$ sequence. To validate the pharmacological behaviour of the $A_3AR$ expression construct, we designed constructs that include the above modifications in a pcDNA5 vector for expression in mammalian cells. The adenosine-bound G protein complex was formed using a dominant negative form of $G\alpha_{i1}$(DNG$\alpha_{i1}$), a dual expression vector containing $G\gamma_2$ and 8xHis-tagged $G\beta_1$[34,66,67]. Four $G\alpha_{i1}$ subunit mutations in DNG$_{i1}$ alter nucleotide binding and affinity for $G\beta\gamma$ to prevent complex dissociation. The Piclidenoson-bound G protein complex was formed using a mini-$G_{si}$ construct (mG$_{si}$), mini-Gs/i148[68]. To facilitate the potential use of scFv16, we replaced the αN of mini-Gs/i148 with $G\alpha_{i1}$ residues G2HN.02 to K35S1.03. mG$_{si}$ was fused to the carboxyl terminus of $A_3AR$ via a GGGS linker[112,116]. Finally, a 3 C cleavage site was incorporated next to the linker prior to the mG$_{si}$ to enable cleavage of the G protein after complex formation. An 8xHis-tagged Nb35 in pET20b was provided by B. Kobilka[69].

## Cell membrane preparation

WT-$A_3AR$ and expression $A_3AR$ constructs were transfected using 1.3 µg DNA per 1 mL of FreeStyle 293 F suspension cells at a ratio of 4:1 PEI/DNA and then incubated for 24 h in a humidified incubator at 37 °C. Transfected cells were harvested and washed with PBS, followed by resuspension in a homogenisation buffer containing 20 mM HEPES, 6 mM MgCl2, 100 mM NaCl, 1 mM EGTA, 1 mM EDTA, pH 7.4. Cell pellets were again resuspended in the homogenisation buffer after centrifugation at 300 g for 3 min at 4 °C. Resuspended cell mixtures were applied to a homogeniser with three bursts for 10 s on ice. Cell nuclei were removed by centrifugation at 600 g for 10 min at 4 °C. The resulting supernatants were further centrifuged at high speed (30,000 × *g*, 60 mins, 4 °C). Cell membranes were collected from pellets and resuspended in a homogenization buffer. The protein concentrations were determined using a bicinchoninic acid (BCA) protein assay. Cell membranes were stored at −80 °C.

## [$^{35}$S]GTPγS binding assay

Adenosine-5′-N-ethyluronamide (NECA)-mediated binding of [$^{35}$S] GTPγS was used to measure G-protein activation by the receptors in membranes extracted from transfected FreeStyle HEK293F cells. Experiments were performed as described previously in ref. 34.

## NanoBRET binding assay

HEK293a cells were seeded into 10 cm dishes at a density of 4 million cells per dish. Following an 8-h incubation, cells were transfected with 5 µg of N-terminal Nluc-tagged $A_3AR$ constructs using polyethylenimine (PEI) at a 4:1 PEI-to-DNA ratio. After 24 h, the cells were replated into 96-well, white-bottom, poly-D-lysine-coated culture plates at a density of 40,000 cells per well. NanoBRET saturation binding was performed as previously reported in refs. 51,52, using the fluorescent $A_3AR$ antagonist CA200645 (final concentration ranging

from 15 nM to 1000 nM). Before the assay, the cell media was replaced with BRET buffer (HBSS supplemented with 10 mM HEPES, 1 unit/mL of ADA, and 10 µM MRS1220 was used to define nonspecific binding. Following equilibration at 37 °C in a humidified atmosphere for 50 min, the substrate 10 µM furimazine or coelenterazine-h was added to each well and cells were incubated for a further 10 min at 37 °C in a humidified atmosphere. Bioluminescence emission wavelengths were measured as previously described[117] using a PheraSTAR Omega plate reader (BMG Labtech) using 460 nm (80 nm bandpass; donor NanoLuc emission) and >610 nm (long pass filter; fluorescent ligand emission). The raw BRET ratio was calculated by dividing the >610 nm emission by the 460 nm emission. All experiments were performed in duplicate. Competitive binding experiments were performed similarly, using a concentration of CA200645 near the $K_d$ (200 nM) of the $A_3AR$ construct and 10-fold serial dilutions of competitors, with an equilibrium time of 50 min. To determine the irreversible binding between LUF7602 and Nluc-tagged hA$_3$AR-BRIL-S97R, we assessed the competitive binding capacity of LUF7602, PSB11, and MRS1220 to two groups of cells (washed and unwashed) at a concentration of 10 µM with 4 h of incubation at room temperature. The washed group was subjected to 4 washing steps with BRET buffer (10 min intervals) to remove unbound ligands. Subsequently, 100 nM of CA200645 was added to both groups and incubated for 50 min before adding furimazine. In the site-directed mutagenesis study, 200 nM of CA200645 was used.

## Trupath assay

HEK293a cells were seeded into 96-well, white-bottom, poly-D-lysine-coated culture plates at a density of 20,000 cells per well. After 4 h of incubation, cells were transiently transfected with equal parts of N-terminal Nluc-tagged $A_3AR$ constructs and G proteins (Gi1-Rluc8, Gβ3, and Gγ9-GFP2), using 15 ng of each DNA construct for a total of 60 ng of DNA per well. PEI was added at a 6:1 PEI-to-DNA ratio. Two days post-transfection, plates were washed twice with BRET buffer (HBSS with 10 mM HEPES, pH 7.4). After a 30-min incubation at 37 °C, plates were placed in a PHERAstar plate reader at 37 °C, and four baseline readings were taken using BRET2 filters at 410 ± 80 nm and 515 ± 30 nm. Increasing concentrations of adenosine, NECA, and CF101 were then added, followed by eight additional readings. BRET2 ratios were calculated as the ratio of GFP2 emission (515 ± 30 nm) to RLuc8 emission (410 ± 80 nm).

## Data and statistical analysis

Data were expressed as the mean ± standard error. All binding and Trupath data were analysed using the nonlinear regression curve-fitting programme with Prism 10 (GraphPad, San Diego, CA, USA). For nanoBRET saturation binding experiments, specific binding was calculated by subtracting non-specific binding (defined as binding in the presence of 10 µM MRS1220) from total binding. Specific binding data in each experiment were then normalised to the maximal binding ($B_{max}$) of WT and fit to a one-site specific binding equation: $Y = B_{max}*[X]/(K_d + [X])$. For nanoBRET competition binding experiments, data were normalised to total and non-specific binding responses and fit to a one-site−Fit $K_i$ equation. Trupath's experimental data was baseline-corrected to the initial four baseline reads and then adjusted to the buffer control response. The area under the curve (AUC) for each response was calculated and plotted with a non-linear regression curve (log[agonist] vs response, three-parameter model), using the equation: $Y =$ Bottom + (Top−Bottom)/$(1 + 10^{((LogEC50-X))})$. Here, $X$ represents the log dose of the agonist, $Y$ is the BRET2 AUC response, Top and Bottom are the curve plateaus, and $EC_{50}$ is the agonist concentration yielding a response halfway between the Bottom and Top values. To determine efficacy values, the efficacy of the agonists ($\tau_A$) was separately calculated using the Black−Leff operational model of agonism[79] using $K_A$ values from Table 2 and then corrected for receptor expression relative

to WT using $B_{max}$ values from Table 1[80]. Significant differences vs WT were determined by one-way ANOVA with a Dunnett's multiple comparison test. Significance levels are indicated as follows: $P \leq 0.0001$****, $P = 0.0001$–$0.001$***, $P = 0.001$–$0.01$**, and $P = 0.01$–$0.05$*.

### Receptor and G protein expression

The Bac-to-Bac Baculovirus Expression System (Invitrogen) produced high-titre recombinant baculovirus for A$_3$AR constructs and DNGα$_{i1}$. BestBac linearised DNA (Expression Systems) was used to make baculovirus for Gβ$_1$γ$_2$. To make the adenosine-bound A$_3$AR-G$_i$ complex, *Trichoplusia ni* (Tni) cells were co-infected with A$_3$AR, DNGα$_{i1}$, and Gβ$_1$γ$_2$ baculovirus at a multiplicity of infection (MOI) ratio of 4:2:1 at a density of 3.5–4 million/mL. To make the Piclidenoson-bound A$_3$AR-mG$_{si}$ complex, a 2:1 ratio of A$_3$AR-mG$_{si}$:Gβ$_1$γ$_2$ baculovirus was used. The A$_3$AR-BRIL-S97R construct was expressed in *S. frugiperda* (Sf9) cells. Insect cell cultures were shaken at 27 °C in ESF 921 serum-free media.

### Expression and purification of scFv16 and Nb35

ScFv16 was expressed and purified as previously described in ref. 65. The expression and purification of Nb35 were adapted from prior methods[69]. In brief, Nb35 was expressed in BL21(DE3) Rosetta *E. Coli* strain using the autoinduction method, followed by purification as previously described.

### Purification of A$_3$AR-G protein complexes

The purification of A$_3$AR-G protein complexes was performed as previously described[34] with minor modifications. The TNI cell pellet was thawed and lysed in a hypotonic buffer containing 20 mM HEPES pH 7.4, 50 mM NaCl, 2 mM MgCl2, 10 µM agonist, protease inhibitors (200 µM phenylmethylsulfonyl fluoride (PMSF), 1 mM leupeptin and trypsin inhibitors (LT), 0.2 mg/mL benzamidine), 1 mg/mL iodoacetamide, 50 µg/mL, 2.5 units of apyrase (NEB) and benzonase (Merk Millipore) and stirred at 25 °C for 20 mins to obtain homogeneity. The cell lysate was centrifuged at $20,000 \times g$ for 20 min at 4 °C. Cell membranes were resuspended and homogenised with a dounce homogenizer in buffer containing 30 mM HEPES pH 7.4, 100 mM NaCl, benzonase (Merk Millipore, 2 uL/800 mL), 2.5 units of apyrase (NEB), protease inhibitors (200 µM PMSF, 1 mM LT, 0.195 mg/mL benzamidine), 2 mM CaCl$_2$, 2 mM MgCl2, and 10 µM agonist in the presence of 0.5% (*w/v*) lauryl maltose-neopentyl glycol (LMNG) buffer 0.03% (*w/v*) cholesterol hemisuccinate (Anatrace, CHS) for 2 h at 4 °C. After removal of insoluble debris by centrifugation ($30,000 \times g$ for 30 min, 4 °C), the supernatant was then loaded onto a glass column filled with M1 anti-Flag antibody resin and washed with 20 column volumes (CV) of wash buffer containing 30 mM HEPES pH 7.5, 100 mM NaCl, 2 mM CaCl$_2$, 2 mM MgCl2, 10 µM agonist, 0.01% (*w/v*) LMNG, and 0.001% (*w/v*) CHS. The receptor complex was eluted from the anti-FLAG resin using the wash buffer supplemented with 0.2 mg/mL flag peptide and 10 mM EGTA. Excessive scFv16 (for DNGα$_{i1}$) or Nb35 (for mGsi) was added to the eluate at a 1:1.5 molar ratio for 30 min at 4 °C. The receptor complex was then concentrated using an Amicon Ultra Centrifugal Filter Unit (MWCO, 100 kDa), filtered (0.22 µm), and subjected to size-exclusion chromatography (SEC) using a Superdex S200 increase 10/300 column (GE Healthcare) in buffer containing 30 mM HEPES pH 7.5, 100 mM NaCl, 10 µM agonist, 0.01% LMNG and 0.001% CHS. Monodisperse fractions of all components were pooled together and spiked with 10 µM agonist before concentration and flash-freezing in small aliquots in liquid nitrogen, and then stored at −80 °C. Protein complexes were confirmed by Coomassie-stained SDS-PAGE gels, detection of FLAG and His epitopes by Western Blot, and by negative stain EM.

### Expression and purification of BAG2, elbow Nb, A$_3$AR-BRIL-S97R, and complex assembly

The BAG2, anti-BRIL Fab fragment, and elbow Nb were expressed in *E. Coli* BL21 (Gold) and purified as previously described in ref. 40.

Purification of A$_3$AR-BRIL-S97R was performed in the absence of ligands with ADA added to remove endogenous adenosine. The purification of A$_3$AR-BRIL-S97R was similar to that of previous methods. Expressed cell pellets were lysed by osmotic shock followed by solubilization in detergent buffer (1% DDM and 0.03% CHS). Solubilised receptor was batch-bound to Ni-chelating resin, followed by washing, and elution. The eluted sample was mixed with anti-Flag M1 antibody resin and loaded onto a glass column. The receptor was buffer-exchanged into 0.1% LMNG detergent with 0.01% CHS and then eluted. The receptor was concentrated and purified further by SEC. Purified A$_3$AR-BRIL-S97R was then incubated with 30 µM LUF7602 and excess BAG2 and elbow Nb at a 1:2:4 molar ratio for 2 h at room temperature and overnight at 4 °C. The LUF7602-bound A$_3$AR-BRIL-S97R-BAG2-Nb complex was purified by SEC. Fractions containing the full complex were collected and spiked with LUF7602 at a final concentration of 10 µM, then incubated on ice for 1 h before being concentrated to 12 mg/mL. The protein complex was confirmed by Coomassie-stained SDS-PAGE gels, detection of FLAG and His tags by Western Blot, and by negative stain EM.

### Cryo-EM sample preparation and data collection

Adenosine-bound A$_3$AR-DNG$_i$-scFv16 cryo-EM grids were prepared on UltrAuFoil 300 mesh 1.2/1.3 grids (Quantifoil, Au300-R1.2/1.3). A$_3$AR-mG$_{si}$-Nb35-Piclidenoson and A$_3$BRIL-S97R-BAG2-Nb-LUF7602 cryo-EM grids were prepared on EMAsian-TiNi 200 mesh 1.2/1.3 grids. Grids were glow-discharged for 180 s (UltrAuFoil grids) or 90 s (EMAsian) at 15 mA current using Pelco EasyGlow in low-pressure air. 3 µL of the purified samples were applied to each grid. Excess sample on the grids was removed by blotting on an FEI Vitrobot Mark IV (Thermo Fisher Scientific) at 100 % humidity and 4 °C with a Whatman #1 filter paper, with a blot force of 12 and blot time of 2 s for UltrAuFoil grids and a blot force of 4 for blot time of 2 s for the EMAsian grids. Blotted grids were then vitrified by plunging into liquid ethane and then liquid nitrogen.

For the adenosine-bound A$_3$AR sample, cryo-EM imaging was performed on the Arctica microscope (Thermo Fisher Scientific) operating at 200 kV equipped with a Gatan K2 Summit direct electron detector in counting mode, corresponding to a pixel size of 1.03 Å. A total of 8676 movies were collected using beam shift with the 9-hole acquisition. Cryo-EM imaging of A$_3$AR-mGs$_i$-Nb35-Piclidenoson was performed on a Titan Krios G3i transmission electron microscope (Thermo Fisher Scientific) operated at an accelerating voltage of 300 kV at a nominal magnification of 105000 in nanoprobe EFTEM mode. Gatan K3 direct electron detector and a 50 µm C$_2$ aperture without an objective aperture inserted during the data collection period were applied to acquire dose-fractionated images of the samples with a slit width of 10 eV, pixel size 0.82 Å. For each movie stack, 60 frames were recorded with a total exposure dose of 60 e$^-$/A$^2$. In total, 6526 movies were recorded in super-resolution mode using SerialEM. For A$_3$BRIL-S97R-BAG2-Nb-LUF7602, 6398 movies were collected on the Titan at a magnification of 130,000× in nanoprobe TEM mode, with electron counting mode with a physical pixel size of 0.65 Å/pixel, exposure rate of 10.57 counts per pixel per second, exposure time of 2.68 s and a total dose of 60 e /Å$^2$.

### Image processing

Data processing for the A$_3$AR-DNG$_{i1}$-scFv16-adenosine complex was performed with the movie stacks subjected to beam-induced motion correction using UCSF MotionCor2[118] using $5 \times 5$ patches. Contrast transfer function (CTF) parameters for each corrected micrograph were calculated by Gctf[119]. Images with Gctf maximum resolution estimates worse than 4 Å were excluded from further processing. Automated particle selection yielded 7,274,091 particles using the Gautomatch software package. The particles were imported into Relion 3.1 for binned extraction with a box size of 240 pixels and

rescaled to 60 pixels. The extracted particles were carried to the cryoSPARC (v3.1) software package[120] and subjected to 2D classification and ab initio 3D and 3D refinement to select good particles, yielding ~1.2 M particles. Particles were processed in Relion3.1[121] for Bayesian particle polishing, CTF refinement, 3D auto-refinement, and subsequent 3D refinement in cryoSPARC, producing a final map at 2.9 Å resolution from 325,569 particles. The resulting particle set was then subjected to 3D refinement and post-processed with a mask excluding the detergent micelle and G proteins, resulting in the final receptor-focused map at a resolution of 3.44 Å, as determined by the FSC 0.143 criteria. The local resolution was estimated using the cryoSPARC v2.15 local resolution estimation function.

For the A3AR–mGsi–Nb35–Piclidenoson complex, all of the processing was performed in cryoSPARC. A total of 6526 micrographs were motion corrected using Patch motion correction, and CTF parameters were estimated using patch CTF estimation. Images with the highest resolution of less than 4 Å were selected for further processing. A total of 5876 movies were finally chosen for particle picking. Particles were first picked using the template picker, followed by 2D classification. Good 2D class averages with randomised orientations produced a particle stack of 2,407,599 particles (from a total of 6.0 M extracted particles). The good particles and 203,651 particles that appeared to be bad were used to generate two ab initio 3D models. The initial models were then used as templates for the initial 3D refinement, followed by another round of 2D classification based on particle set matching features of the GPCR heterotrimeric complex. The subsequent selected 2D averages, containing 943,812 particles, were applied to another round of 2D classification and hetero refinement using the same initial models, resulting in a final particle set for further processing. Finally, 589,921 particles were selected for a non-uniform 3D consensus refinement and CTF envelope fitting, yielding a map with a resolution of 2.6 Å at a Fourier shell correlation of 0.143 (gold standard). The dataset was subjected to further local-motion correction (per particle), and the final resolution was improved to 2.5 Å. Subsequently, to further improve resolution in the transmembrane region, a mask excluding the detergent micelle and G proteins was implemented to calculate a high-quality map of the receptor, resulting in a receptor-focused density map of 3.1 Å resolution, which was used to build the receptor model.

For the A3BRIL–S97R–BAG2–Nb–LUF7602 complex, movies were motion-corrected using Patch Motion Correction in CryoSPARC (v3.1). Patch CTF was used to estimate defocus values. Micrographs with CTF resolution estimates worse than 4 Å were excluded from further examination. The ~3.3 M particles were auto-picked using a previously generated template. The particles were extracted with a box size of 360 pixels and rescaled to 90 pixels. The extracted particles were subjected to 2D classification, ab initio 3D, and heterogeneous 3D refinement. Particles were re-extracted at full resolution and imported into cryoSPARC3.1 for Bayesian particle polishing and CTF refinement. The shiny particles were imported into the CryoSPARC (v3.1) for additional 2D and 3D classifications. The final set of 328,104 yielded a final map at 2.6 Å. For A3AR, a local class for the receptor-only region was performed with a mask excluding the BRIL-BAG2-Nb to improve the map quality for the ligand binding pocket, thereby assisting in modelling. This resulted in the final consensus map at 3.3 Å, as determined by the FSC 0.143 criteria.

## Model building and refinement

**A3AR–DNGi1–scFv16–adenosine complex modelling.** An initial homology model of A3AR was generated using the SWISS-MODEL server[122] with the activated structure of A1AR (PDB: 7LD4) as a template[36]. Models of Gi heterotrimers and scFv16 were adopted from the dopamine receptor D3R-Gi-Pramipexole complex (PDB: 7CMU)[123]. The model was docked into the cryo-EM map using the "fit in map" routine in UCSF Chimera[124]. This starting model was then subjected to

rigid body fitting and followed by iterative rounds of automated refinement in Phenix real-space refinement[125] and repeated rounds of manual building in Coot[126].

**A3AR–mGsi–Nb35–Piclidenoson complex modelling.** The receptor from the A3AR–DNGi1–scFv16–adenosine model was rigid-body placed into the receptor-focused cryo-EM map. The mGsi, Gβ1γ2, and Nb35 were modified from the adhesion GPCR ADGRD1 structure (PDB: 7WU2)[127] and were rigid-body placed in the consensus map. The fitted models of all subunits were further refined through several iterations of manual model building in COOT and real-space refinement, as implemented within the Phenix software.

**A3AR–BRIL–S97R–BAG2–Nb–LUF7602 complex.** A homology model of A3AR generated by SWISS-MODEL using the A1AR crystal structure (5UEN)[32] as a template was placed into the receptor-focused map in ChimeraX, followed by repeated rounds of model building in Coot and iterative refinement with Phenix. The BRIL-BAG2-Nb trimer module was obtained from the cryo-EM structure of the Frizzled complex (PDB:6WW2)[39] and was docked into the consensus cryo-EM density map using ChimeraX and rigid-body placed with Phenix. Refinements with Phenix and manual building in Coot were performed on the trimer. To aid with molecular modelling, composite maps were made from DeepEMhancer post-processing cryo-EM maps of the consensus and receptor-focused refinement maps[128]. The molecular model was refined against the composite map (EMD-48065). All the final models were visually inspected for their general fit to the map and validated using Molprobity[129]. Structural figures were prepared using ChimeraX[130]. The cryo-EM data collection, refinement and validation report for this complex are shown in Table S1.

## IFD

The IFD protocol of Schrödinger was used to dock XAC into A3AR. This protocol uses a combination of Glide XP and Prime to accurately model ligand and protein side chains. The protein was prepared using the Schrödinger Protein Preparation Wizard. This process adds hydrogen atoms, assigns bond orders, optimises the hydrogen-bonding network, and performs a restrained minimisation using the OPLS4 force field. XAC was prepared from SMILES using Schrödinger LigPrep, to generate 3D structures, different ionisation states and tautomeric forms at a physiological pH of 7.4. The ligand structures were minimised using the OPLS4 force field. The grid box was built with the centroid of the C59 ligand, Prime refinement of side chains within 5 Å, and a maximum of 20 poses were redocked with Glide XP.

## Reporting summary

Further information on research design is available in the Nature Portfolio Reporting Summary linked to this article.

## Data availability

Atomic coordinates for the A3AR–BRIL–BAG2-Nb–LUF7602, A3AR–DNGi1–scFv16–adenosine, and A3AR–mGsi–Nb35–Piclidenoson complex structures were deposited in the Protein Data Bank (PDB) with the accession codes 9EHS, 9EBH, and 9EBI, respectively. Cryo-EM maps were deposited in the Electron Microscopy Data Bank under the accession codes EMD-47879 (consensus map) and EMD-47994 (receptor focus map) for the A3AR–DNGi1–scFv16–adenosine complex, EMD-47880 (consensus map) and EMD-47998 (receptor focus map) for the A3AR–mGsi–Nb35–Piclidenoson complex, and EMD-48063 (consensus map) and 48064 (receptor focus map), EMD-48065 (composite map) for the A3AR–BRIL–BAG2-Nb–LUF7602 complex. Atomic coordinates for previously determined structures can be accessed via accession codes: 5UEN, 5MZP, 5N2S, 5MZJ, 8GNG, 3REY, 7LD4, 6GDG, 8HDP, 2YDO, 9EBH, 9EBI, 8YH2. The source data underlying Figs. 2G, H, I, J; 6A–C, F; and 7F, G and Supplementary

Figs. 1B; 2B–E, G, H; 6A, and 7B are provided as a Source Data file. Source data are provided with this paper.

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

## Acknowledgements

The AI tool Claude 3.5 Sonnet (Anthropic, 2024) was used for proof-reading the manuscript and verifying consistency in spelling, grammar, and text clarity. It was not used for generating ideas, content, figures, or data. This work was funded by an Australian Research Council (ARC) Linkage Project LP180100560 (DMT), National Health and Medical Research Council of Australia (NHMRC) project grant 1138448 (DMT),

and NHMRC Investigator Grant APP1196951 (DMT). A.G. is a CSL Centenary Fellow. L.T.M. is supported by a National Heart Foundation Future Leader Fellowship (101857). This work was partially supported by the Monash University Ramaciotti Centre for cryo-electron microscopy and the Monash University MASSIVE high-performance computing facility.

## Author contributions

L.Z., L.T.M., A.C., A.G., and D.M.T. conceived the study. L.Z., J.I.M., A.T.N., F.M.B., L.T.M., A.G., and D.M.T. designed the experiments. D.E. and L.H.H. provided chemical tools. L.Z. and F.M.B. performed the pharmacology experiments. L.Z. performed the biochemistry experiments. L.Z., J.I.M., H.V., and A.G. performed sample vitrification and cryo-EM imaging. L.Z., J.I.M., H.V., A.G., and D.M.T. processed the EM data and generated and analysed atomic models. D.M.T., L.T.M., and A.G. acquired funding for this project. A.T.N., L.H.H., J.I.M., L.T.M., A.G., and D.M.T. provided project supervision. L.Z., J.I.M., and D.M.T. wrote the initial manuscript draft with reviewing and editing from all authors.

## Competing interests

A.C. is a co-founder and shareholder of Septerna Inc. The remaining authors declare no competing interests.
