## [Transparent Peer Review file · Nature Communications]

Molecular basis of ligand binding and receptor activation at the human A₃ adenosine receptor

Corresponding Author: Dr David Thal

Version 0:

Reviewer comments:

Reviewer #1

(Remarks to the Author)

The manuscript investigates the molecular basis of ligand binding and receptor activation at the A₃AR, implicated in various physiological and pathological processes. Using cryo-EM, mutagenesis, and pharmacological assays, the authors present structures of the A₃AR in three distinct functional states: inactive (bound to the covalent antagonist LUF7602), active (bound to the endogenous agonist adenosine), and activated by the clinically relevant agonist Piclidenoson. The study uncovers unique structural features and activation mechanisms that enhance our understanding of A₃AR pharmacology and provide a framework for designing selective therapeutics targeting A₃AR for diseases such as inflammatory conditions, cancer, and glaucoma. Since numerous structures of the adenosine receptor in both active and inactive states have already been reported, the structures presented in this paper offer limited novelty. Additionally, the manuscript appears to have been prepared hastily.

1. While the inactive structure of A₃AR and the covalent attachment of the ligand to Y265 are key contributions of this manuscript, the density visualization in the figures should be enhanced to better display the ligand density, as the provided map lack density to support the covalent bond. Moreover, the authors should provide additional evidence, such as mass spectrometry or other analytical approaches, to substantiate the covalent attachment between the Y265 side chain and LUF7602.
2. The authors analyzed interactions between A₃AR and LUF7602/Piclidenoson, the author should provide a figure showing the density of the critical amino acids involved in the binding pocket and indicate the contour level in the figure legend.
3. Both LUF7602 and Piclidenoson are selective compounds against A₃AR, it may be necessary to strengthen the explanation of A₃AR's selectivity mechanism.
4. Many mutagenesis experiments are included in this manuscript, data on the expression levels of the mutated constructs are missing. It would be helpful to provide information on the key mutations to confirm that these mutations do not notably impact expression levels.
5. Two articles have previously reported the active structure of A₃AR. A parallel comparison between the authors' analyzed A₃AR structure and these two published structures would be valuable.
6. "disulphide bond" should be "disulfide bond".
7. The calculation symbols in the tables and supplementary data are not displaying correctly.
8. The authors should describe the details of the engineered mini-Gsi construct in the Methods section.
9. The manuscript lacks line numbers, which makes it necessary to refer to specific sections or context to locate text.

Reviewer #2

(Remarks to the Author)

This work describes the cryo-EM structural determination of the A3AR in complex with two agonists (both native agonist adenosine and Phase 3 molecule piclidenoson) and one antagonist. One high affinity reversible antagonist was initially used – but the resolution was insufficient, so the team used an irreversibly binding, xanthine-based ligand that was previously reported, LUF7602. This allowed more effective resolution of the ECLs and other features than when a reversibly bound agonist was present in previous structures. Each portion of the receptor is carefully analyzed and compared with the structures of the other three adenosine receptor subtypes. The findings go significantly beyond the content of several recent structures of the A3AR (published in two papers cited appropriately as refs. 37 and 38), that were only agonist-bound structures and were lacking essential information about extracellular loop structures, which are essential for the selectivity. The inactive structures were made possible using an anti-BRIL Fab + nanobody, and some mutations – and enable conclusions about receptor activation that is mediated by a H-bond network. For example, the putative rotamer toggle switch Trp residue in TM6 has moved upon activation, leading to TM6 outward movement.

This study finally explains the interaction of the iodo group of the 3-iodobenzyl moiety of piclidenoson: “the iodine atom points towards the backbone carbonyl of M172(45.56)” to establish a halogen bond.

The large outward movement of TM2 in the LUF7602 complex is striking. Is this a necessary conformational change to inactivate the receptor? What does this movement do to the G protein contact region, if anything, on the cytosolic side? How is this reconciled with the proposed outward movement of TM2 (based on modeling, but not experimental structures) when some C2 extended potent agonists are bound to the A3 receptor (see refs. 108 and 111)? In that case the proposed outward movement does not preclude receptor activation.

The ligand dependent effects on efficacy compared to binding affinity are new and useful, especially the contrast between difference agonists, as shown in Figure 6. The most unexpected change is the affinity increase for the V169E mutation. That is left without explanation. Are there any potential H-bonding groups in vicinity? Has a halo bond with the 3-iodo group been considered? Molecular dynamics simulation might clarify the basis of this affinity enhancement. Concerning the statement on p. 8, “...consistent with prior observations that higher affinity agonists may have lower signalling efficacy (79).” This reference concerns muscarinic receptors – not adenosine receptors. This phenomenon is not a general observation for A3 agonists. For A3 agonists, affinity and maximal efficacy are entirely independent parameters.

The following line: “the N250A(6.55(and H272A(7.43) mutants did not signal suggesting these mutations impair receptor function.” If the impairment is only on binding affinity, this statement is unjustified. If there is no agonist binding, nothing can be concluded about the receptor activation function.

Table S2 has the same data in the second half as in the first half (but labeled Emax, instead of pEC50). Please correct. Also, the column headings show (n), which would indicated number of determinations – but those numbers are missing.

Minor comments

Page 2, line 10 of 2nd paragraph: Correct A1AR – with only “1” as subscript. This type of error has been propagated in the past leading to confusion.

Page 3, [35S]-GTPγS should be without a hyphen.

Page 6, 6 lines before the section “Activation mechanism”, a word is missing: ... our docking of to the A3AR
The parenthesis after the statement “in an orientation” on p. 3 is unreadable on the pdf: sigma1 etc.

Version 1:

Reviewer comments:

Reviewer #1

(Remarks to the Author)

The revised manuscript by Zhang and coauthors shows improvement and addresses my previous concerns. I think the current version is suitable for publication in Nature Communications.

Reviewer #2

(Remarks to the Author)

All of this reviewer's concerns have been appropriately answered.

Title: Molecular basis of ligand binding and receptor activation at the human A3 adenosine receptor

Manuscript: NCOMMS-24-83675A

The revised manuscript by Zhang and coauthors shows improvement and addresses my previous concerns. I think the current version is suitable for publication in Nature Communications.

REVIEWER COMMENTS

We thank both reviewers for taking the time to provide useful and warranted feedback on our manuscript. We have made corrections as discussed below.

Reviewer #1 (Remarks to the Author):

The manuscript investigates the molecular basis of ligand binding and receptor activation at the A₃AR, implicated in various physiological and pathological processes. Using cryo-EM, mutagenesis, and pharmacological assays, the authors present structures of the A₃AR in three distinct functional states: inactive (bound to the covalent antagonist LUF7602), active (bound to the endogenous agonist adenosine), and activated by the clinically relevant agonist Piclidenoson. The study uncovers unique structural features and activation mechanisms that enhance our understanding of A₃AR pharmacology and provide a framework for designing selective therapeutics targeting A₃AR for diseases such as inflammatory conditions, cancer, and glaucoma. Since numerous structures of the adenosine receptor in both active and inactive states have already been reported, the structures presented in this paper offer limited novelty. Additionally, the manuscript appears to have been prepared hastily.

1. While the inactive structure of A₃AR and the covalent attachment of the ligand to Y265 are key contributions of this manuscript, the density visualization in the figures should be enhanced to better display the ligand density, as the provided map lack density to support the covalent bond.

In Figure 1C, we show the density surrounding LUF7602, which includes a lack of density supporting the covalent bond. We have zoomed in further in Figure 2A to highlight this better as well.

Figure 1C

Figure 2A

Moreover, the authors should provide additional evidence, such as mass spectrometry or other analytical approaches, to substantiate the covalent attachment between the Y265 side chain and LUF7602.

In previous work, wash-resistant pharmacology was observed for LUF7602 (17b) in a radioligand binding assay (manuscript reference #39). A Y265F mutation abolished the wash-resistant pharmacology, confirming covalent attachment to Y265 as the mechanism. We show similar wash-resistant pharmacology for LUF7602 in a NanoBRET binding assay (Supplementary Figure 2G).

Similarly, from (ref #37), LUF7602 (17b) displays wash-resistant inhibition of $[^3H]$ PSB-11 binding.

The mutation $A_3AR-Y265F$ displays no wash-resistant inhibition of $[^3H]$ PSB-11 binding.

We have clarified these points further in the main text, lines 141-148:

“We note that we did not observe clear EM density for the covalent linkage between the benzene-sulfonate group of LUF7602 and Y265^{7.36}. We assigned the covalent attachment based on our pharmacology experiments showing irreversible binding (Figure S2G), a prior study identifying Y265^{7.36} as the point of covalent attachment (39), and similarity to the A1AR bound to the irreversible antagonist DU172 (PDB: 5UEN) (32). Poor EM density around the benzene-sulfonate group, Y265^{7.36}, and Y15^{1.35} (Figure 2A) suggests conformational heterogeneity potentially due to the covalent attachment not being 100% complete consistent with ~15% rebinding of XAC-630 in the washout experiments (Figure S2G).”

2. The authors analysed interactions between A3AR and LUF7602/Piclidenoson, the author should provide a figure showing the density of the critical amino acids involved in the binding pocket and indicate the contour level in the figure legend.

As suggested, we have updated panels in Figure 2A and 5B and included the contour level in the figure legends.

Figure 2A

Figure 5B

3. Both LUF7602 and Piclidenoson are selective compounds against A3AR, it may be

necessary to strengthen the explanation of A3AR's selectivity mechanism.

We have expanded our discussion of mechanisms of selectivity for A₃AR agonists and antagonists, which includes more discussion of prior work (which partially addresses comment #5 as well).

See lines 462–501.

4. Many mutagenesis experiments are included in this manuscript, data on the expression levels of the mutated constructs are missing. It would be helpful to provide information on the key mutations to confirm that these mutations do not notably impact expression levels.

Expression data (B_{max}) were reported in the original submission as mentioned in lines 198-206, shown in Figure 2G, and reported in Table 1. B_{max} values were used to correct log τ used to values in Table 3.

5. Two articles have previously reported the active structure of A3AR. A parallel comparison between the authors' analyzed A3AR structure and these two published structures would be valuable.

Supplemental Figure 10 was created that compares structures across these studies in more detail. We note that differences in the resolution of the structures limit the utility of some of these comparisons. For example, we compared Piclidenoson's binding mode to the prior Piclidenoson-bound A₃AR structure (PDB: 8X16). The position of the N⁶-iodobenzene group differs between the structures (Figure S8), with residue M174^{5,35} playing a key role in this difference (lines 330-347). However, the poor EM map density around the N⁶-iodobenzene group and nearby residues in 8X16 suggest potential uncertainties in the side chain and ligand modelling. We have added the more recent structure with Namodenoson-bound to the sheep A₃AR structure (PDB: 8YH6) to this section (Figure S9), noting similar concerns about the EM density of the N⁶-iodobenzene group and nearby residues in 8YH6. We feel these are important distinctions to make in the era of AlphaFold3 and cryo-EM, where high-resolution structural details of ligand-binding interactions remain crucial for understanding mechanism (PMID: 39643640).

6. “disulphide bond” should be “disulfide bond”.

The spelling of disulphide bond is consistent with the Oxford English Dictionary used by the Nature Springer journals, which includes Nature Communications.

7. The calculation symbols in the tables and supplementary data are not displaying correctly.

We apologise for this PDF conversation error, which was remedied in the revised manuscript.

8. The authors should describe the details of the engineered mini-Gsi construct in the Methods section.

Details of our engineered mini-Gsi was edited for clarity on lines 601-606.

“The Piclidenoson-bound G protein complex was formed using a mini-Gsi construct (mGsi), mini-Gs/i148 (68). To facilitate the potential use of scFv16 we replaced the αN of mini-Gs/i148 with Gαi1 residues G2HN.02 to K35S1.03. mGsi was fused to the carboxyl terminus of A₃AR via a GGS linker (112, 116). Finally, a 3C cleavage site was incorporated next to the linker prior to the mGsi to enable cleavage of the G protein after complex formation.

9. The manuscript lacks line numbers, so text must be located by referring to specific sections or context.

Line numbers have been added.

Reviewer #2 (Remarks to the Author):

This work describes the cryo-EM structural determination of the A₃AR in complex with two agonists (both native agonist adenosine and Phase 3 molecule piclidenoson) and one antagonist. One high affinity reversible antagonist was initially used – but the resolution was insufficient, so the team used an irreversibly binding, xanthine-based ligand that was previously reported, LUF7602. This allowed more effective resolution of the ECLs and other features than when a reversibly bound agonist was present in previous structures. Each portion of the receptor is carefully analyzed and compared with the structures of the other three adenosine receptor subtypes.

The findings go significantly beyond the content of several recent structures of the A₃AR (published in two papers cited appropriately as refs. 37 and 38), that were only agonist-bound structures and were lacking essential information about extracellular loop structures, which are essential for the selectivity.

The inactive structures were made possible using an anti-BRIL Fab + nanobody, and some mutations – and enable conclusions about receptor activation that is mediated by a H-bond network. For example, the putative rotamer toggle switch Trp residue in TM6 has moved upon activation, leading to TM6 outward movement.

This study finally explains the interaction of the iodo group of the 3-iodobenzyl moiety of piclidenoson: “the iodine atom points towards the backbone carbonyl of M172(45.56)” to establish a halogen bond.

We thank Reviewer #2 for highlighting how our study goes beyond several recent A₃AR structures.

The large outward movement of TM2 in the LUF7602 complex is striking. Is this a necessary conformational change to inactivate the receptor?

The large outward movement of the TM2 in the extracellular region of the LUF7602 structure is due to the presence of the large benzene-sulfonate linkage of LUF7602 (Figure 1E). It creates a steric clash with V72-S73 on TM2. This results in a $\sim 3\text{\AA}$ outwards shift of the top of TM2. This movement propagates to $\sim D58^{2,50}$. This shift is similar and actually less than the one created by DU172 and the A₁AR (3\AA at the A₃AR vs 5\AA at the A₁AR; lines 167-170). It is unlikely that this conformational change is related to receptor activation but is, rather, due to ligand binding. The largest TM2 shift was observed in the MRS1220-A₃AR structure (8\AA), but due to its low resolution and the lack of clarity regarding whether it is ligand-bound, we have limited our discussion on this structure (lines 172-176).

What does this movement do to the G protein contact region, if anything, on the cytosolic side?

There are differences between the ICL1 in the LUF7602 and piclidenoson/adenosine-bound structures, but these are most likely caused by the presence of G protein in the active-state structures. The TM2 shift due to LUF binding is lost at $\sim D58^{2,50}$.

How is this reconciled with the proposed outward movement of TM2 (based on modeling, but not experimental structures) when some C² extended potent agonists are bound to the A₃ receptor (see refs. 108 and 111)? In that case the proposed outward movement does not preclude receptor activation.

Our hypothesis is that the top of TM2 movement is related to ligand binding and is not tied to receptor activation. This model is in agreement with receptor activation by larger and bulkier C² extended agonists.

We thank the reviewer for highlighting these points, and have added them to the discussion on ligand selectivity (lines 470-481).

“Subtype selective A₃AR agonists have been designed by introducing modifications at the N⁶ and C² positions of adenosine. Modifications at the C² position, such as the Cl group in Namodenoson, extend towards TM2 (Fig. S10C). Similarly, the A₃AR selective antagonist LUF7602 and the A₁AR selective antagonist DU172 both contain moieties that extend towards TM2. Our comparison of inactive and active A₃AR structures revealed an outward movement of TM2 at the extracellular side of the receptor. This conformational change appears to be associated with ligand binding rather than receptor activation, as evidenced by the convergence of TM2 conformations near D58^{2,50} in both states. Supporting this interpretation, molecular modeling studies with bulkier C²-extended agonists (90, 91) predicted similar TM2 movements. These observations suggest that TM2 flexibility accommodates diverse C² substituents and contributes to ligand selectivity without directly participating in receptor activation mechanisms.”

The ligand dependent effects on efficacy compared to binding affinity are new and useful, especially the contrast between difference agonists, as shown in Figure 6. The most unexpected change is the affinity increase for the V169E mutation. That is left without explanation. Are there any potential H-bonding groups in vicinity? Has a halo bond with the 3-iodo group been considered? Molecular dynamics simulation might clarify the basis of this affinity enhancement.

It turns out the increase in affinity for piclidenoson at the V169E mutant was observed in a prior study that included MD simulations of the V169E mutant, which we now appropriately cite (PMID: 31502843). The results of that study suggest the increase was due to increased hydrophobic interactions with TM5 and TM6. Another possibility would be an anion-aromatic interaction between the Glu and the edge of the 3-iodo group (PMID: 23798413).

We have added the following explanation to the manuscript (lines 326-331), noting that we also corrected the increase in binding affinity from 10-fold to 5-fold:

“However, the mutation resulted in a 5-fold increase in binding affinity for Piclidenoson (**Figure 6F**). Similar results for the V169E^{45,53} mutant were observed in a prior study that included examination of the V169E^{45,53} mutant in molecular dynamic simulations (74). The simulations suggest increased hydrophobic interactions with M174^{5,35}, M177^{5,38}, and I253^{6,58}. Another possible explanation for the increase in affinity could be due to the creation of an anion-aromatic interaction (75) between E169 and the edge of the N⁶-iodobenzene group.”

Concerning the statement on p. 8, “...consistent with prior observations that higher affinity agonists may have lower signalling efficacy (79).” This reference concerns muscarinic receptors – not adenosine receptors. This phenomenon is not a general observation for A3 agonists. For A3 agonists, affinity and maximal efficacy are entirely independent parameters.

We thank the reviewer for reminding us of this point, as we have removed this statement (line 385).

The following line: “the N250A(6.55(and H272A(7.43) mutants did not signal suggesting these mutations impair receptor function.” If the impairment is only on binding affinity, this statement is unjustified. If there is no agonist binding, nothing can be concluded about the receptor activation function.

We appreciate this point as well and have removed this statement (line 386).

Table S2 has the same data in the second half as in the first half (but labeled Emax, instead of pEC50). Please correct. Also, the column headings show (n), which would indicated number of determinations – but those numbers are missing.

Thanks you for pointing out this error, we have corrected Table S2 to reflect the data in Figure 7F.

Minor comments

Page 2, line 10 of 2nd paragraph: Correct A1AR – with only “1” as subscript. This type of error has been propagated in the past leading to confusion.

We resolved this typo (line 79).

Page 3, [35S]-GTPyS should be without a hyphen.

We removed the hyphen for all occurrences of [35S]-GTPyS.

Page 6, 6 lines before the section “Activation mechanism”, a word is missing: ... our docking of to the A3AR.

We edited this sentence for clarity (line 343)

The parenthesis after the statement “in an orientation” on p. 3 is unreadable on the pdf: sigma1 etc.

We apologise for this PDF conversation error.

Response to Reviewers

We thank the reviewers for their time and effort in reviewing our manuscript.

Reviewer #1 (Remarks to the Author):

The revised manuscript by Zhang and coauthors shows improvement and addresses my previous concerns. I think the current version is suitable for publication in Nature Communications.

Reviewer #2 (Remarks to the Author):

All of this reviewer's concerns have been appropriately answered.